# The autophagy initiator ULK1 sensitizes AMPK to allosteric drugs

Toby A. Dite[1], Naomi X.Y. Ling[1], John W. Scott[2,3], Ashfaqul Hoque[1], Sandra Galic[2], Benjamin L. Parker[4], Kevin R.W. Ngoei[2], Christopher G. Langendorf[2], Matthew T. O'Brien[2], Mondira Kundu[5], Benoit Viollet [6,7,8], Gregory R. Steinberg[9], Kei Sakamoto[10,11], Bruce E. Kemp [2,3] & Jonathan S. Oakhill[1,3]

AMP-activated protein kinase (AMPK) is a metabolic stress-sensing enzyme responsible for maintaining cellular energy homeostasis. Activation of AMPK by salicylate and the thieno-pyridone A-769662 is critically dependent on phosphorylation of Ser108 in the β1 regulatory subunit. Here, we show a possible role for Ser108 phosphorylation in cell cycle regulation and promotion of pro-survival pathways in response to energy stress. We identify the autophagy initiator Unc-51-like kinase 1 (ULK1) as a β1-Ser108 kinase in cells. Cellular β1-Ser108 phosphorylation by ULK1 was dependent on AMPK β-subunit myristoylation, metabolic stress associated with elevated AMP/ATP ratio, and the intrinsic energy sensing capacity of AMPK; features consistent with an AMP-induced myristoyl switch mechanism. We further demonstrate cellular AMPK signaling independent of activation loop Thr172 phosphorylation, providing potential insight into physiological roles for Ser108 phosphorylation. These findings uncover new mechanisms by which AMPK could potentially maintain cellular energy homeostasis independently of Thr172 phosphorylation.

[1] Metabolic Signalling Laboratory, St Vincent's Institute of Medical Research, University of Melbourne, Melbourne, VIC, Australia. [2] Protein Chemistry & Metabolism, St Vincent's Institute of Medical Research, University of Melbourne, Melbourne, VIC, Australia. [3] Mary MacKillop Institute for Health Research, Australian Catholic University, Melbourne, VIC, Australia. [4] Charles Perkins Centre, School of Molecular Bioscience, The University of Sydney, Sydney, NSW, Australia. [5] Department of Pathology, St Jude Children's Research Hospital, Memphis, TN, USA. [6] INSERM, U1016, Institut Cochin, Paris, France. [7] CNRS, UMR8104, Paris, France. [8] Université Paris Descartes, Sorbonne Paris Cité, Paris, France. [9] Divisions of Endocrinology and Metabolism, Department of Medicine, and Department of Biochemistry and Biomedical Sciences, McMaster University, Hamilton, ON, Canada. [10] MRC Protein Phosphorylation and Ubiquitylation Unit, School of Life Sciences, University of Dundee, Scotland, UK. [11] Present address: Nestlé Institute of Health Sciences SA, Lausanne, Switzerland. Toby A. Dite and Naomi X.Y. Ling contributed equally to this work. Correspondence and requests for materials should be addressed to J.S.O. (email: joakhill@svi.edu.au)

The evolutionarily conserved AMP-activated protein kinase (AMPK) is a key regulator of cellular and whole-body energy homeostasis that controls multiple branches of metabolism to redress energy imbalances caused by physiological and pathological processes[1, 2]. AMPK senses increased cellular AMP/ATP ratio during periods of energy stress (hypoxia, nutrient deprivation, exercise) and protects the cell from these events by switching off energy-consuming anabolic pathways and switching on catabolic pathways to restore ATP levels. Multiple physiological processes are regulated by AMPK including autophagy, appetite control, mitochondrial biogenesis and cell growth, and proliferation. Consequently, extensive efforts have been made to develop AMPK-activating drugs for potential therapeutic use in treating metabolic diseases (type 2 diabetes, obesity, cardiovascular disease) and also cancer and inflammatory diseases.

The AMPK αβγ heterotrimer comprises an α-catalytic subunit and regulatory β- and γ-subunits. Multiple isoforms of each subunit exist in mammals (α1/2, β1/2, γ1/2/3) and isoform-specific variations in tissue distribution, regulation, and function have been demonstrated. Both β-isoforms contain a carbohydrate-binding module (CBM) and are myristoylated at position Gly2, a modification that targets AMPK to intracellular membranes and is important for temporospatial regulation of AMPK signaling[3, 4]. γ-subunits possess three allosteric adenylate nucleotide-binding sites that bind ATP, ADP and AMP interchangeably, enabling AMPK to sense fluctuations in cellular energy state[5–7].

In most instances, ligand-induced allosteric regulation of AMPK is governed by distinct phosphorylation events that either sensitize AMPK to nucleotides/drugs binding at γ-subunit sites (phosphorylation of Thr172 (pThr172) in the α-subunit activation loop[8–10]), or small compounds binding at the ADaM (allosteric drug and metabolism) site (phosphorylation of β-Ser108 (pSer108) in the β-CBM[11–13]). An exception is synergistic activation of unphosphorylated AMPK when γ- and ADaM sites are occupied simultaneously[10, 13]. The ADaM site, a largely hydrophobic cavity formed between the α-kinase domain small lobe and β-subunit CBM, was identified in crystal structures of

AMPK/drug complexes[12, 14]. These structures revealed that the phosphate group of pSer108 forms electrostatic interactions with α2-kinase domain residues Thr21, Lys31, and Lys33, thereby stabilizing the ADaM site and explaining the role of pSer108 in mediating drug sensitization. The precise mechanism by which ADaM site drugs activate AMPK is not fully understood. pSer108 is absolutely required for AMPK activation by A-769662[11, 13], salicylate[15] and MT47-100[16], and increases by 40-fold potency of the high affinity drug 991[12]. Using purified enzyme we further demonstrated that activation loop phosphorylation is dispensable for AMPK stimulation by A-769662[14]. The well-defined character of the AMPK drug site, and its regulation through reversible phosphorylation, has led to speculation that synthetic activators (991, A-769662) and salicylate are mimicking an endogenous metabolite(s) that would be capable of sustaining AMPK signaling in the absence of pThr172[1, 17].

Thr172 phosphorylation is considered a marker of AMPK activity; identification of upstream kinases, and the mechanisms underpinning pThr172 regulation, have been the subject of intense investigation over several decades. LKB1 and CaMKK2 (Ca$^{2+}$/calmodulin-dependent protein kinase kinase 2) have been identified as in vivo Thr172 kinases[18]. Despite some conflicting evidence, current models of AMPK regulation by adenine nucleotides describe a tripartite mechanism in which ATP exchange for AMP and ADP at γ-sites (i) promotes Thr172 phosphorylation, (ii) suppresses pThr172 dephosphorylation, and (iii) (for AMP) allosterically activates Thr172-phosphorylated AMPK[3, 5, 6]. Hierarchical phosphorylation events in the α-subunit Ser/Thr rich ST-loop (human α1(472–525)) have also been reported to negatively regulate pThr172, either by suppressing Thr172 phosphorylation (α-Ser487 auto-, Akt- or PKA-phosphorylation[1, 2]), or by promoting pThr172 dephosphorylation (α-Thr479 phosphorylation by GSK3[19]). Activity of the autophagy initiator Unc-51-like kinase (ULK), itself an AMPK substrate, is associated with reduced pThr172 via an uncharacterized negative feedback loop[20].

In contrast, regulation and function of β-Ser108 phosphorylation have been largely unexplored. Ser108 is highly conserved

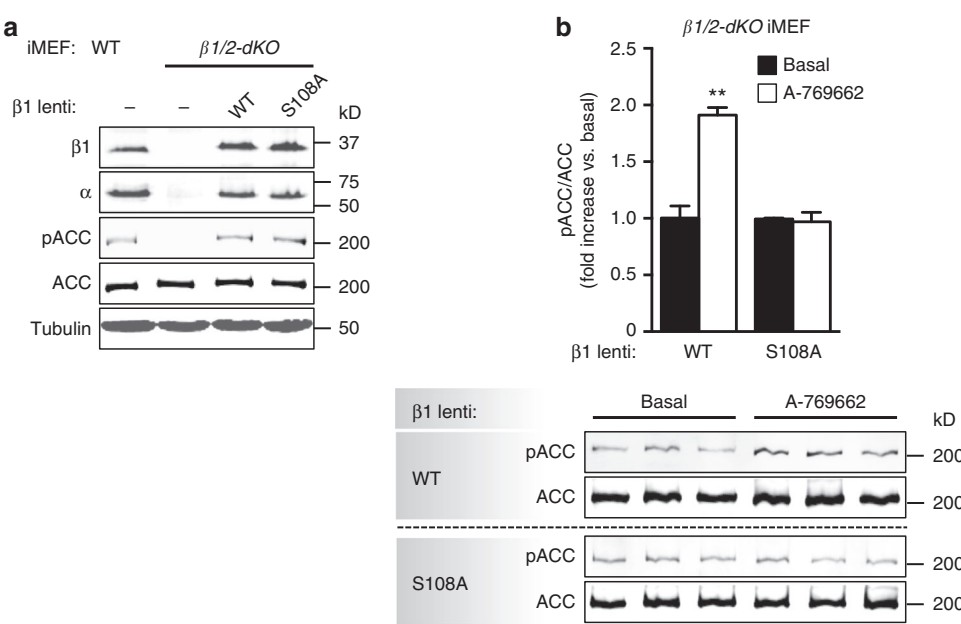

**Fig. 1** A-769662 activation of cellular AMPK signaling is dependent on β1-pSer108. **a** Reconstitution of basal AMPK signaling in AMPK β1/2 double knockout (*β1/2-dKO*) iMEFs by lentiviral transduction of AMPK β1 WT or S108A mutant. **b** Immunoblots for pACC from *β1/2-dKO* iMEFs-expressing β1 WT or S108A mutant, stimulated with 20 μM A-769662 for 90 min. *n = 3*. *Error bars*, mean pACC fold change relative to basal ± s.e.m. Statistical analysis was performed using unpaired two-tailed Student's *t*-test. \*\*P < 0.01 indicates significant increase in pACC compared to basal

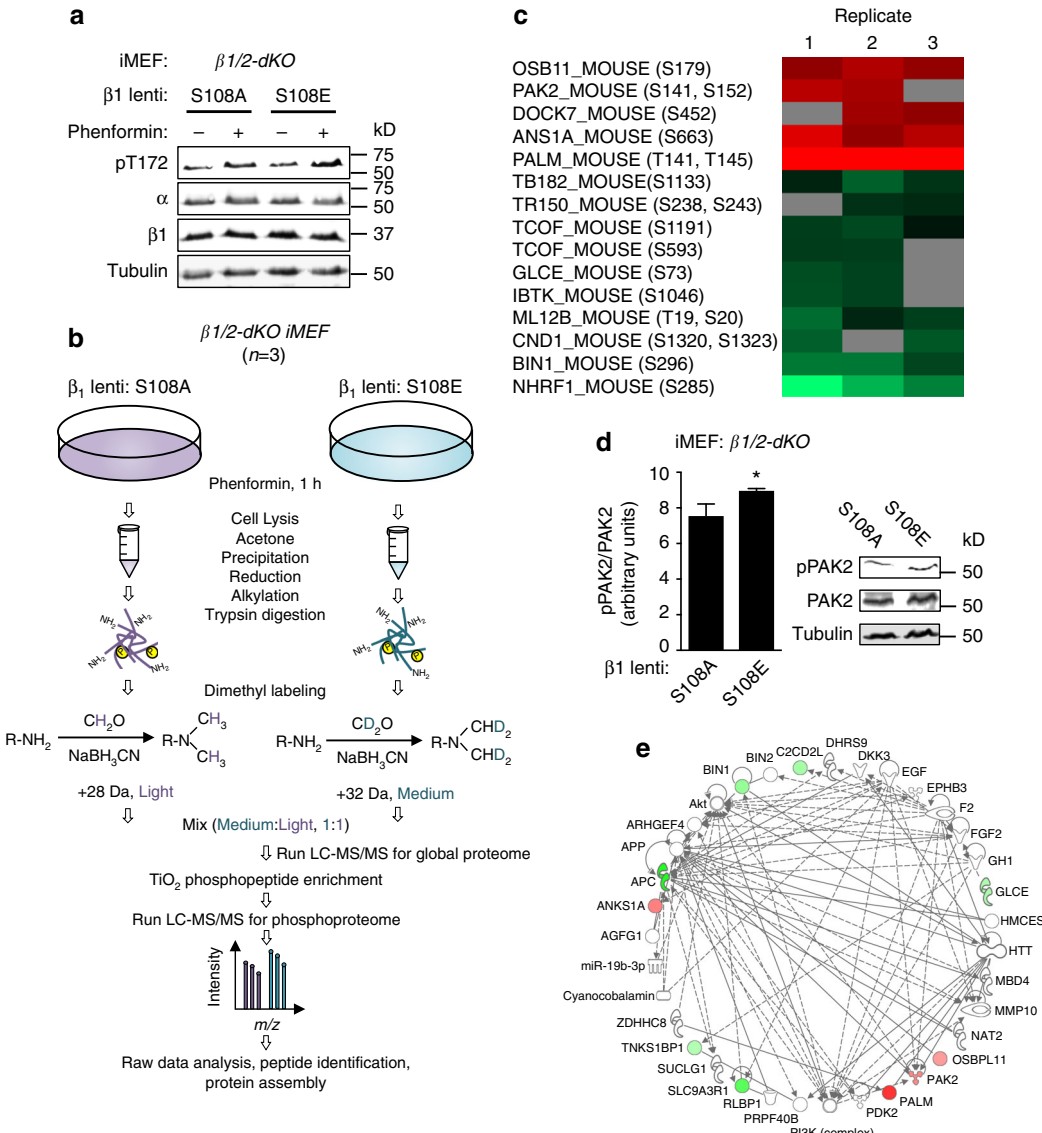

**Fig. 2** Quantitative global and phosphoproteomic analysis uncovers cellular roles for β1-pSer108. **a** Representative immunoblots for *β1/2-dKO* iMEFs-expressing β1 mutants S108A or S108E, stimulated with 2 mM phenformin for 1 h. **b** Workflow showing the stable isotope dimethyl labeling-based quantitative proteomic and phosphoproteomic approach. **c** Heatmap showing significantly perturbed cellular phosphoproteins and corresponding phosphosites. *Red* indicates increased, and *green* decreased, phosphorylation in β1-S108E compared to β1-S108A-expressing cells. *Gray* indicates missing phosphopeptide in that replicate. **d** Immunoblot/densitometry analysis confirming increased PAK2-Ser141 phosphorylation in β1-S108E-expressing cells. *n* = 3, representative immunoblot is shown. *Error bars*, mean PAK2-Ser141 phosphorylation (arbitrary units) ± s.e.m. Statistical analysis was performed using unpaired two-tailed Student's *t*-test. *P < 0.05 indicates significant increase in PAK2-pSer141 in S108E-expressing cells compared to S108A-expressing cells. **e** ''Cell cycle, connective tissue development and function, cellular movement'' is one of the top networks associated with changes in phosphoproteome between S108A and S108E-expressing iMEFs, as identified by Ingenuity Pathway Analysis software

in eukaryotes and was identified as an autophosphorylation site in rat liver AMPK preparations[21]. Closer examination using kinase inactive (KI) AMPK expressed in COS-7 cells revealed that Ser108 is a *cis*-autophosphorylation site (dependent on intramolecular Thr172 phosphorylation) that is dephosphorylated following removal of the AMPK-activating stimulus. Thus, as for Thr172, Ser108 is largely unphosphorylated under basal conditions[13]. Identification of alternate upstream kinases for Ser108 would provide advances in two important areas: firstly, characterization of novel Ser108 kinases implicates their therapeutic modulation as a strategy to increase potency of AMPK-targeting drugs; secondly, feed-forward regulation by Ser108 kinases might denote the cellular processes under which concentrations of natural AMPK ligands become elevated. Here, we demonstrate

that AMPK β1-Ser108 is a substrate for ULK1 under conditions associated with elevated AMP. We also provide unambiguous demonstration of AMPK signaling independently of Thr172 phosphorylation. These findings underpin a ULK-mediated "ligand switch" model of AMPK allosteric control, in which the adenylate charge-sensing role of β1-AMPK is replaced by an ability to detect perturbations in endogenous metabolite(s) acting at the ADaM site.

## Results

**β1-pSer108 confers cellular AMPK drug sensitivity.** The importance of Ser108 phosphorylation in sensitizing AMPK to ADaM site ligands has been well-characterized using purified

AMPK enzyme in vitro, but whether this translates to cellular AMPK signaling has not been unequivocally demonstrated. To exclude signaling from endogenous AMPK we generated an immortalized mouse embryonic fibroblast (iMEF) cell line derived from mouse embryos harboring genetic deletion of both β1 and β2 isoforms[22] (*β1/2-dKO*). As expected, *β1/2-dKO* iMEFs were devoid of detectable AMPK α- or β-subunit expression and AMPK signaling as evidenced by lack of phosphorylation of the AMPK substrate ACC-Ser79 (Fig. 1a). Lentiviral-transduction of either FLAG-tagged wild-type (WT) or S108A mutant β1 in *β1/2-dKO* iMEFs reconstituted expression of AMP-sensitive AMPK heterotrimers, and recovered phenformin-sensitive AMPK signaling in transduced cells (Fig. 1a and Supplementary Fig. 1a, b). Incubation with the direct AMPK agonist A-769662 (20 μM) led to a significant increase (1.9-fold) in pACC-Ser79 in *β1/2-dKO* iMEFs-expressing WT β1, but not the β1 S108A mutant (Fig. 1b). A-769662-stimulation was mediated exclusively through the ADaM site, since phosphorylation of Thr172 was not increased at this dose (Supplementary Fig. 1c). These results confirm a requirement for Ser108 phosphorylation in drug activation of AMPK in cells.

**Phosphoproteomic analysis hints at roles for β1-pSer108.** To investigate the cellular fate of β1-Ser108 phosphorylation, we performed a stable isotope dimethyl labeled-based quantitative proteomic and phosphoproteomic analysis using *β1/2-dKO* iMEFs, transduced with either β1 mutant S108A or S108E (Fig. 2a). We previously showed that Glu at position 108 acts as an effective phosphomimetic, with regard sensitizing AMPK to activation by A-769662[13]. Following 1 h phenformin (2 mM) treatment, β1 S108A and S108E transduced iMEF lysates were compared to trace out the changes in proteome and phospho-proteome (Fig. 2b). Fifteen cellular phosphoproteins showed significant changes in phosphorylation between S108A and S108E-expressing cells in our study (Fig. 2c, Supplementary Table 1) with no detectable differences in AMPK expression or phenformin-induced Thr172 phosphorylation (Fig. 2a, Supplementary Fig. 2a). For example, we identified increased phosphorylation of p21-activated kinase 2 (PAK2) on Ser141/Ser152, located in the kinase inhibitory domain, in phenformin-treated iMEFs-expressing β1 S108E, whereas global proteome data showed no changes in PAK2 protein level (Supplementary Fig. 2b, c). Immunoblot analysis with a phosphospecific antibody confirmed significant increase in pPAK2-Ser141 in S108E-expressing cells, validating our representative phosphoproteome data (Fig. 2d). Collectively, pathway and network analysis using ingenuity pathway analysis (IPA) identified "Cell cycle, connective tissue development and function, cellular movement" as

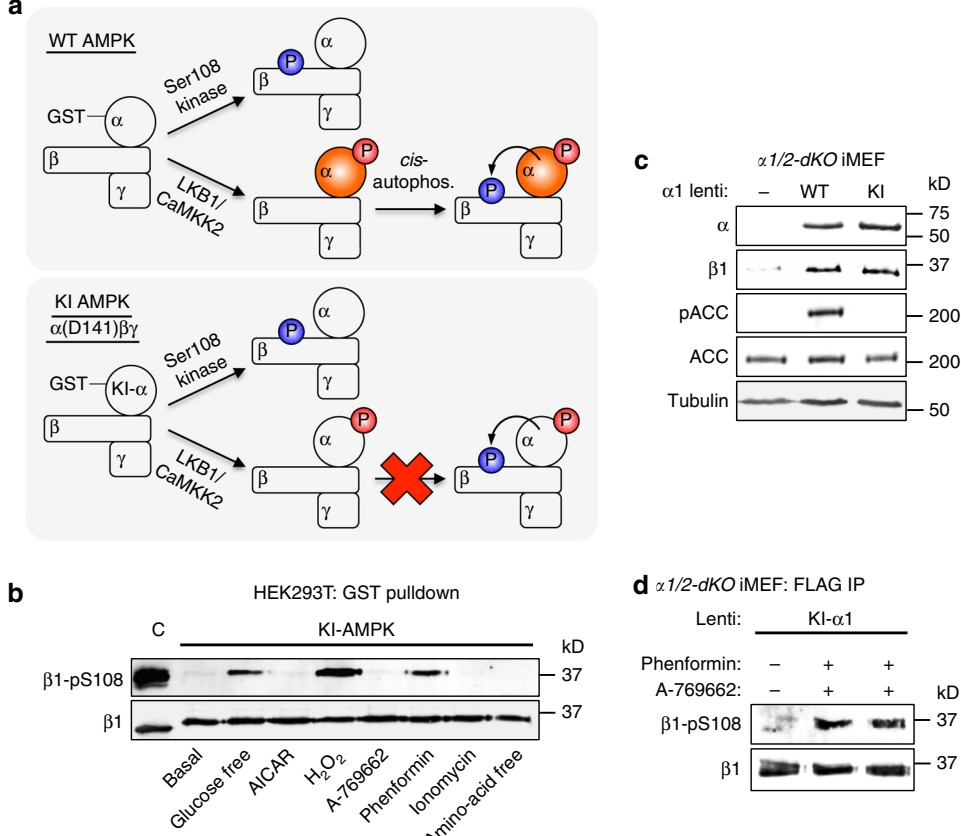

**Fig. 3** β1-Ser108 *trans*-phosphorylation occurs via an AMPK independent mechanism. **a** Rationale for employing kinase inactive (KI) AMPK to examine cellular Ser108 phosphorylation. β1-Ser108 phosphorylation (*blue*) can potentially be performed by an upstream kinase in both WT and KI α1(D141A) AMPK mutant. LKB1/CaMKK2-mediated phosphorylation of α-Thr172 (*red*) activates WT AMPK (*orange*) leading to background Ser108 *cis*-autophosphorylation. This is excluded using KI AMPK, which can be phosphorylated on Thr172 but remains inactive. **b** Immunoblot for β1-pSer108 in KI-α1β1γ1 purified from HEK293T cells treated with AMPK-activating agents/conditions: glucose free (glucose-free DMEM + 10% serum, 4 h), AICAR (2 mM, 1 h), $H_2O_2$ (1 mM, 45 min), A-769662 (300 μM, 1 h), phenformin (2 mM, 1 h), ionomycin (2.5 μM, 15 min) and amino-acid free (EBSS medium, 4 h). *n* = 3, representative immunoblots shown. **C**: Bacterial expressed, CaMKK2-treated α1β1γ1 standard. **c** Reconstitution of basal AMPK signaling in AMPK α1/2 double knockout (α1/2-dKO) iMEFs by lentiviral transduction of AMPK α1 WT, but not the KI mutant. **d** Immunoblot for β1-pSer108 from α1/2-dKO iMEFs-expressing KI-α1, stimulated with 2 mM phenformin for 1 h. *n* = 3, representative immunoblots shown

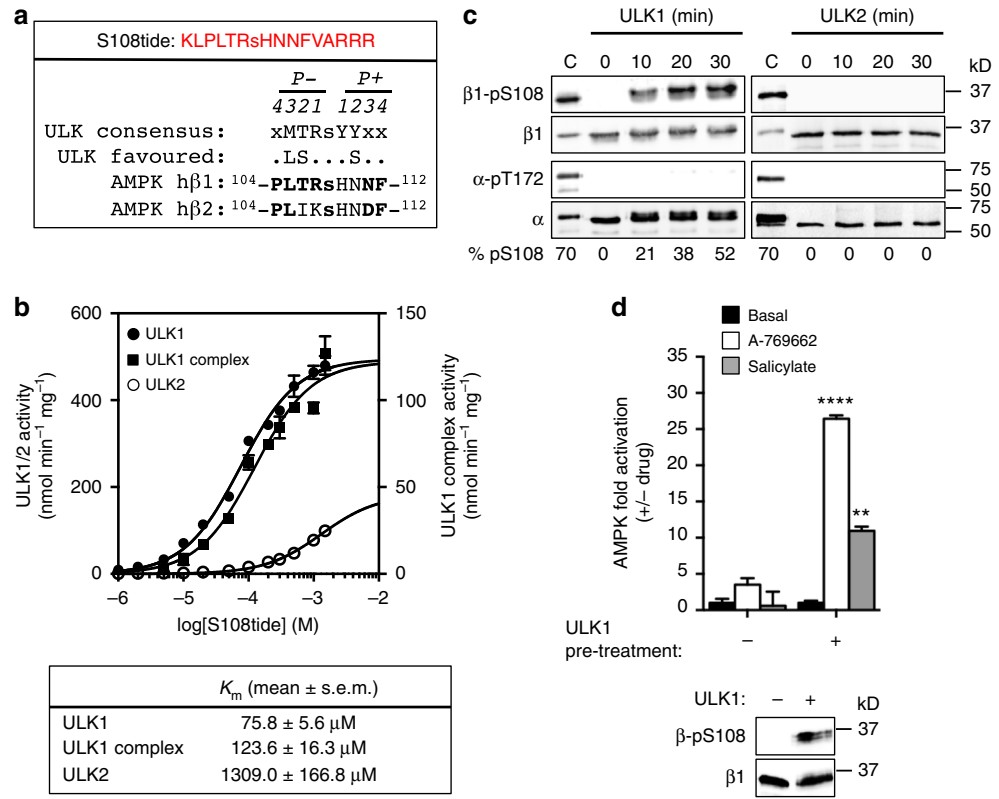

**Fig. 4** ULK1 phosphorylates β1-Ser108 in vitro. **a** Sequence alignment of the ULK consensus motif/favorable substitutions[23] with human AMPK β1- and β2-residues 104–112 (Ser108 in lower case). x denotes positions with no demonstrated preference. Preferred/favored consensus β-residues are in *bold*. Sequence of the synthetic peptide S108tide is shown in *red*. **b** Dose curve of S108tide phosphorylation by ULK1 and ULK2 (plotted to left y-axis), and ULK1/FIP200/Atg13 complex (plotted to *right* y-axis). $n = 3$. *Error bars*, mean activity ± s.e.m. **c** Immunoblots for β1-pSer108 and α-pThr172 in bacterial-expressed KI-α1β1γ1 phosphorylated with ULK1 (*left*) or ULK2 (*right*) for 30 min. $n = 3$, representative immunoblots shown. C: CaMKK2-treated α1β1γ1 control. **d** Activity of ULK1-phosphorylated α1(C176S)β1γ1 in the presence of 20 μM A-769662 or 10 mM salicylate. $n = 3$, representative immunoblots of pSer108 in assayed AMPK preparations are shown. *Error bars*, mean fold AMPK activation relative to ULK1-untreated ± s.e.m. Statistical analyses were performed using one way ANOVA with post hoc Dunnett's multiple comparison test. **$P < 0.01$, ****$P < 0.0001$ indicate significant increase in AMPK activation compared to ULK1-untreated

one of the top networks associated with these perturbed cellular phosphoproteins (Fig. 2e).

**β1-Ser108 can be phosphorylated in *trans*.** We previously demonstrated that AMPK β1-Ser108 is *cis*-autophosphorylated via an intramolecular mechanism reliant on prior phosphorylation of α-Thr172[13]. This implies that, in the absence of alternate signaling, aberrant phosphatase regulation or conditions leading to *trans*-autophosphorylation, stoichiometries of pSer108 and pThr172 are intrinsically linked and, for the most part, equivalent. To interrogate this model we screened AMPK-activating conditions/agents under which Ser108 becomes a substrate for a *trans*-phosphorylation event in cells. To prevent background *cis*-signaling we expressed a GST-fusion of the KI AMPK mutant α1 (D141A)β1γ1 in HEK293T cells. This complex is a suitable substrate for Thr172-phosphorylating kinases LKB1 and CaMKK2, but does not undergo Ser108 *cis*-autophosphorylation (Fig. 3a). Glucose starvation, or incubation with $H_2O_2$ or phenformin, each induced phosphorylation at Ser108 (Fig. 3b). Incubation with ionomycin (activator of CaMKK2 signaling), AICAR (5-aminoimidazole-4-carboxamide ribonucleotide) or A-769662, or amino-acid deprivation, each failed to elicit Ser108 phosphorylation at the single experimental time point used. We used α1/2-dKO iMEFs[23] to exclude the possibility that Ser108 is *trans*-autophosphorylated in phenformin-treated cells. Lentiviral-transduction of α1/2-dKO iMEFs with FLAG-tagged WT- or

KI-α1 reconstituted AMPK expression; as expected only WT α1 transduction recovered AMPK signaling (Fig. 3c). However, phenformin induced phosphorylation of Ser108 in α1/2-dKO iMEFs transduced with KI-α1, which was not increased by additional incubation with 100 μM A-769662 (Fig. 3d). Our results demonstrate that, under certain metabolic stress conditions, Ser108 is a substrate for a kinase(s) other than AMPK autophosphorylation.

**ULK1 phosphorylation of β1-Ser108 induces drug sensitivity.** To identify upstream kinases for β1-Ser108 we screened a synthetic peptide corresponding to AMPK β1(102–114) (S108tide) (Fig. 4a) against a panel of 284 Ser/Thr kinases. 92% of the kinases screened yielded low activities against S108tide (<10% vs. top hit). Several kinases from diverse groups demonstrated comparable (>35%) specific activity relative to AMPK α1β1γ1, e.g., BRSK1/2, NEK2/9, TAOK1 (Supplementary Table 2). Among positive hits was ULK1, which is in accordance with identity of the sequence surrounding β1-Ser108 to the consensus motif for substrates of this kinase[24], particularly Leu, Thr, and Arg directly N-terminal to Ser108 (positions P-3, P-2, and P-1, respectively) (Fig. 4a). We performed a more detailed analysis and phosphorylated the S108tide substrate using purified FLAG-ULK1, either alone ($k_{cat}/K_m$ $1.23 \times 10^{-2}$/s/μM) or complexed to interaction partners FIP200 and Atg13 ($k_{cat}/K_m$ $1.86 \times 10^{-3}$/s/μM) (Fig. 4b and Supplementary Fig. 3a). ULK2 demonstrated

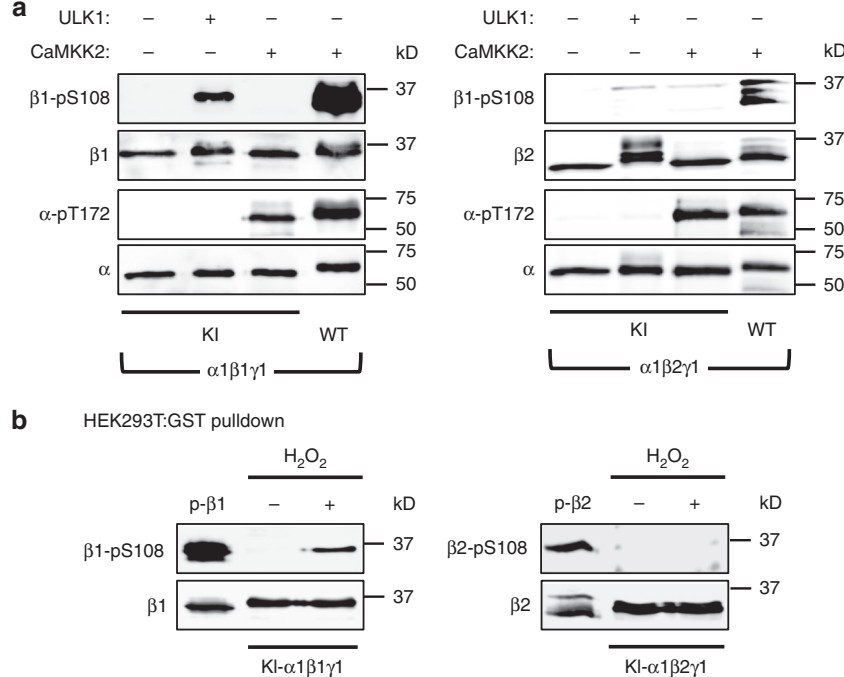

**Fig. 5** ULK1 phosphorylation of Ser108 is specific for the AMPK β1 isoform. **a** Immunoblots for β1-pSer108, β2-pSer108, and α-pThr172 in bacterial-expressed KI-α1β1γ1 (*left*) or KI-α1β2γ1 (*right*) phosphorylated with ULK1 or CaMKK2 for 30 min. Controls: CaMKK2-treated WT α1β1γ1 (*left*) or α1β2γ1 (*right*). **b** Immunoblots for β-pSer108 in KI-α1β1γ1 (*left*) or KI-α1β2γ1 (*right*) purified from HEK293T cells stimulated with 1 mM $H_2O_2$ for 45 min. Controls: p-β1 and p-β2, CaMKK2-treated WT α1β1γ1 and α1β2γ1, respectively. In both panels n = 3, representative immunoblots shown

reduced efficiency in phosphorylating S108tide ($k_{cat}/K_m$ $2.63 \times 10^{-4}$/s/μM), compared to ULK1 (Fig. 4b). We examined ULK phosphorylation of the AMPK heterotrimer using purified KI-α1β1γ1 expressed in bacteria, since we previously found the WT complex extracted from this source is autophosphorylated at β1-Ser108 with ~ 60% stoichiometry despite lacking pThr172[13]. ULK1 phosphorylated β1-Ser108, but not α1-Thr172, in KI-AMPK, whereas we did not detect phosphorylation of either residue by ULK2 (Fig. 4c). MS/MS analysis confirmed ULK1-phosphorylation of Ser108 in KI-α1β1γ1 (Supplementary Fig. 3b, c).

We found that expression of the mutant α1(C176S)β1γ1 in bacteria produced AMPK that was devoid of pSer108 (Supplementary Fig. 3d) yet retained full allosteric regulatory mechanisms, such as A-769662/AMP and A-769662/C2 synergistic activation (Supplementary Fig. 3e)[10, 13], or individual A-769662 (Supplementary Fig. 3f) and AMP (Supplementary Fig. 3g) activation following phosphorylation by CaMKK2. One explanation is that pThr172-independent basal activity, and hence Ser108 autophosphorylation, is induced by modification of Cys176 in the activation loop, occurring as a result of oxidative stress during AMPK expression in bacteria. Redox sensitive mechanisms are known to regulate receptor tyrosine kinases[25] and a variety of Ser/Thr kinases including CaMK2, PKA and PKC[26–28]. The α1(C176S)β1γ1 construct allowed us to investigate the effect of ULK phosphorylation on ligand-mediated regulation of AMPK without the need for prior and extensive phosphatase treatment. We found that ULK pre-treatment sensitized purified AMPK α1(C176S)β1γ1 to activation by A-769662 or salicylate (Fig. 4d). We detected ULK1-phosphorylation at several other AMPK α1, β1, and γ1 sites (Supplementary Fig. 3c). These included γ1 residues Ser261 and Ser270, located in proximity to nucleotide site 3, which is important for AMP allosteric regulation[3, 7]. However, AMP allosteric activation of WT α1β1γ1 was not significantly affected by pre-treatment with ULK1 (Supplementary Fig. 3g).

**ULK1 phosphorylation of β-Ser108 is specific to AMPK β1.** Ser108 is conserved between mammalian AMPK β-isoforms 1 and 2, however identity of the β2 sequence to the ULK consensus/favored motif is restricted to Leu at P-3 (Fig. 4a). ULK1 was previously shown to phosphorylate multiple AMPK β2 residues in vitro, but not Ser108[20]. To confirm β-isoform specificity of ULK1 we generated a phosphospecific antibody to β2-pSer108 (Supplementary Fig. 4). Incubation of WT α1β1γ1 or α1β2γ1 with CaMKK2 resulted in Thr172 phosphorylation and subsequent Ser108 autophosphorylation (Fig. 5a). ULK1 treatment of KI-α1β2γ1 resulted in reduced electrophoretic migration of β2, indicative of multiple phosphorylation events. However, we did not detect β2-pSer108 by immunoblot following ULK1 treatment (Fig. 5a, right panel). Furthermore, exposure of HEK293T cells to $H_2O_2$ failed to induce Ser108 phosphorylation in KI-α1β2γ1 (Fig. 5b, right panel).

**β1-Ser108 is a cellular substrate for ULK.** We examined whether ULK phosphorylates β1-Ser108 in HEK293T cells. Initially, we used the highly selective, ULK1 small molecule inhibitor SBI-0206965 (termed 6965)[24]. We found that pre-incubation of transfected HEK293T cells with 10 μM 6965, a concentration previously shown to have no effect on AMPK signaling[24], significantly reduced $H_2O_2$- (Fig. 6a) and phenformin- (Fig. 6b) induced phosphorylation of Ser108 in KI-AMPK α1β1γ1, relative to 6965 untreated cells. Treatment with 6965 caused a significant increase in pThr172 in KI-α1β1γ1 in response to both $H_2O_2$ and phenformin (Fig. 6a, b); this is consistent with a role for ULK1 as a negative regulator of Thr172 phosphorylation.

We investigated regulation of Ser108 phosphorylation in iMEFs in which both ULK1 and ULK2 had been genetically deleted (*ulk1/2-dKO*)[29] (Supplementary Fig. 5). Under basal conditions, pSer108 in endogenous AMPK was significantly higher in both WT and *ulk1/2-dKO* iMEFs compared to

HEK293T cells, despite no detectable increase in pThr172 (Fig. 6c). We bypassed this high basal signal in iMEFs by examining pSer108 in KI-AMPK, expressed by lentiviral transduction of a FLAG-tagged KI-α1(D141A) mutant and isolated with endogenous β- and γ-subunits by FLAG immunoprecipitation. KI-AMPK expressed in iMEFs was devoid of basal pSer108 (Fig. 6d). Phenformin-induced phosphorylation of Ser108 was significantly decreased (>90%) in KI-AMPK from *ulk1/2-dKO* iMEFs, compared to WT iMEFs (Fig. 6d). Combined, these results confirm β1-Ser108 as a cellular substrate for ULK1.

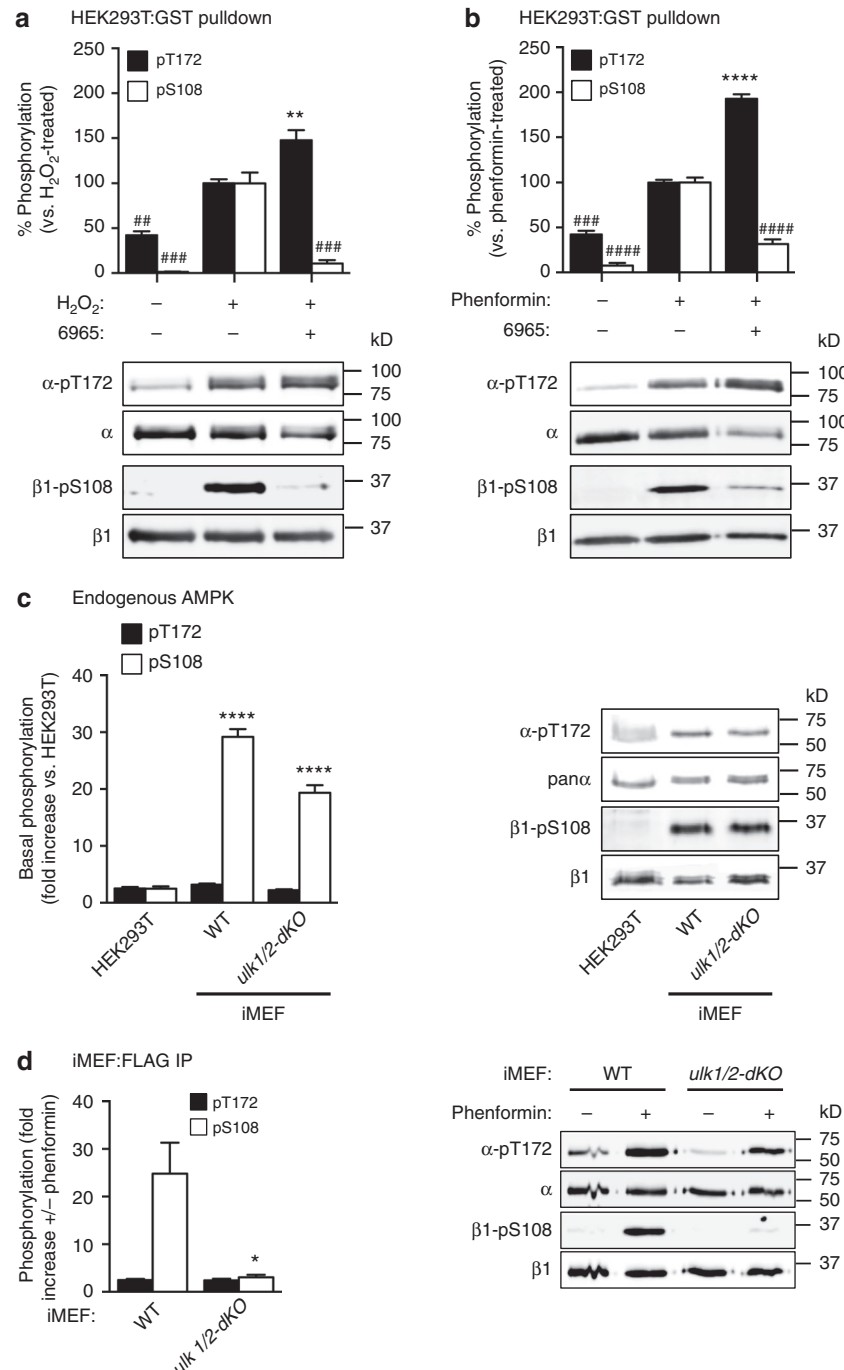

**Fig. 6** ULK phosphorylates β1-Ser108 in cells. Statistical analyses were performed using one-way ANOVA with post hoc Dunnett's multiple comparison test, unless indicated. Immunoblots for β1-pSer108 and α-pThr172 in KI-α1β1γ1 purified from HEK293T cells incubated with **a** 1 mM $H_2O_2$ and 10 μM 6965 for 45 min, or **b** 2 mM phenformin and 10 μM 6965 for 1 h. $n = 3$, representative immunoblots shown. *Error bars*, mean % phosphorylation relative to $H_2O_2$- or phenformin-treated ± s.e.m. **$P < 0.01$ indicates significant increase, and ##$P < 0.01$, ###$P < 0.001$ and ####$P < 0.0001$ indicate significant decrease, compared to $H_2O_2$- or phenformin-treated. **c** Immunoblots for β1-pSer108 and α-pThr172 in lysates from HEK293T cells, WT or *ulk1/2-dKO* iMEFs incubated in 25 mM glucose DMEM + 10% serum. $n = 3$ individual cultures per cell line, representative immunoblots shown. *Error bars*, mean fold increase in phosphorylation relative to HEK293T cells ± s.e.m. ****$P < 0.0001$ indicates significant increase in phosphorylation compared to HEK293T cells. **d** Immunoblots for β1-pSer108 and α-pThr172 in KI-α1 AMPK purified from WT or *ulk1/2-dKO* iMEFs stimulated with 2 mM phenformin for 1 h. $n = 3$, representative immunoblots shown. *Error bars*, mean fold increase in phosphorylation relative to basal ± s.e.m. Statistical analysis was performed using unpaired two-tailed Student's *t*-test. *$P < 0.05$ indicates significant decrease compared to WT iMEFs

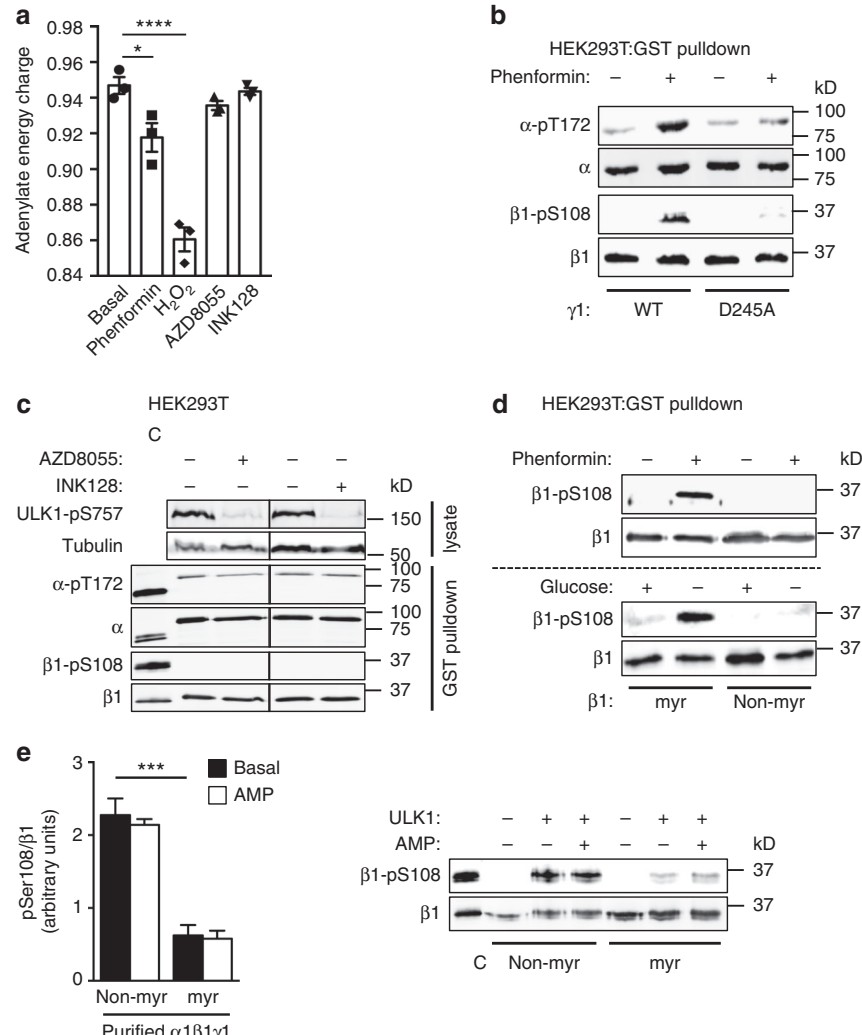

**Fig. 7** An AMP-myristoyl switch triggers ULK1 phosphorylation of β1-Ser108. **a** Adenine nucleotides extracted from HEK293T cells incubated with phenformin (2 mM, 1 h), $H_2O_2$ (1 mM, 45 min), AZD8055 (1 μM, 1 h), or INK128 (1 μM, 1 h) were quantitated by mass spectrometry. Adenylate energy charge was calculated as described in Online Methods. $n = 3$. Error bars, mean adenylate energy charge ± s.e.m. Statistical analyses were performed using one-way ANOVA with post hoc Dunnett's multiple comparison test. ****$P > 0.001$, *$P < 0.05$ indicate significant decrease in mean adenylate energy charge compared to basal. **b** Immunoblots for β1-pSer108 and α-pThr172 in KI-α1β1γ1 or KI-α1β1γ1(D245A) purified from HEK293T cells stimulated with 2 mM phenformin for 1 h. $n = 3$, representative immunoblots shown. **c** Immunoblots for ULK1-pSer757, and β1-pSer108 and α-pThr172 in KI-α1β1γ1 purified from HEK293T cells incubated with 1 μM mTOR inhibitors AZD8055 or INK128 for 1 h. $n = 3$, representative immunoblots shown. C: Bacterial expressed, CaMKK2-treated α1β1γ1 standard. **d** Immunoblots for β1-pSer108 in KI-α1β1γ1 (myr) or KI-α1β1(G2A)γ1 (non-myr) purified from HEK293T cells incubated with 2 mM phenformin for 1 h (upper) or glucose free medium for 4 h (lower). $n = 3$, representative immunoblots shown. **e** Immunoblot for β1-pSer108 in bacterial-expressed, non-myristoylated (non-myr) or myristoylated (myr) KI-α1β1γ1 phosphorylated with ULK1 for 30 min in the presence of 100 μM AMP. $n = 3$, representative immunoblots shown. C: CaMKK2-treated α1β1γ1 standard. Error bars, mean increase in β1-pSer108 relative to ULK1-untreated ± s.e.m. Statistical analysis was performed using unpaired two-tailed Student's t-test. ***$P < 0.001$ indicates significant decrease in β1-pSer108 relative to non-myristoylated AMPK

**An AMP myristoyl switch triggers β1-Ser108 trans-phosphorylation.** $H_2O_2$ and phenformin indirectly activate AMPK through perturbation of adenine nucleotide ratios (increased AMP/ATP ratio, reduced adenylate charge). We found that $H_2O_2$ and phenformin treatments of HEK293T cells that induced Ser108 phosphorylation (Fig. 3b) also produced significant falls in adenylate energy charge (AEC), with $H_2O_2$ incubation having the greater effect (AMP/ATP ratios: basal 0.0093 ± 0.0015; phenformin 0.0177 ± 0.0041; $H_2O_2$ 0.0418 ± 0.0033) (Fig. 7a). Therefore, we investigated whether elevated AMP was a requirement for phosphorylation of Ser108 by ULK1. Phenformin treatment of HEK293T cells failed to induce Ser108 phosphorylation in KI-α1β1γ1 carrying a mutation in the γ1 nucleotide site 3 (γ1-D245A) that renders AMPK insensitive to stimulation by

AMP[3] (Fig. 7b). ULK1 can also be activated in response to small molecule mTOR inhibitors (AZD8055, INK128) that suppress mTOR-mediated phosphorylation at the ULK1 inhibitory site Ser757[24]. Incubation of HEK293T cells with 1 μM AZD8055 or INK128 induced almost complete loss of ULK1-pSer757, without significantly affecting adenylate charge or stimulating phosphorylation of Ser108 in KI-α1β1γ1 expressed in these cells (Fig. 7a, c). These data indicate that ULK1 phosphorylation of β1-Ser108 in cells requires a reduction in adenylate charge and is dependent on the AMP-sensing abilities of AMPK.

N-terminal myristoylation of β-Gly2 plays important roles in AMPK temporospatial regulation, being required for AMP-stimulation of Thr172 phosphorylation by upstream kinases and metabolic stress-induced co-localization of AMPK to both

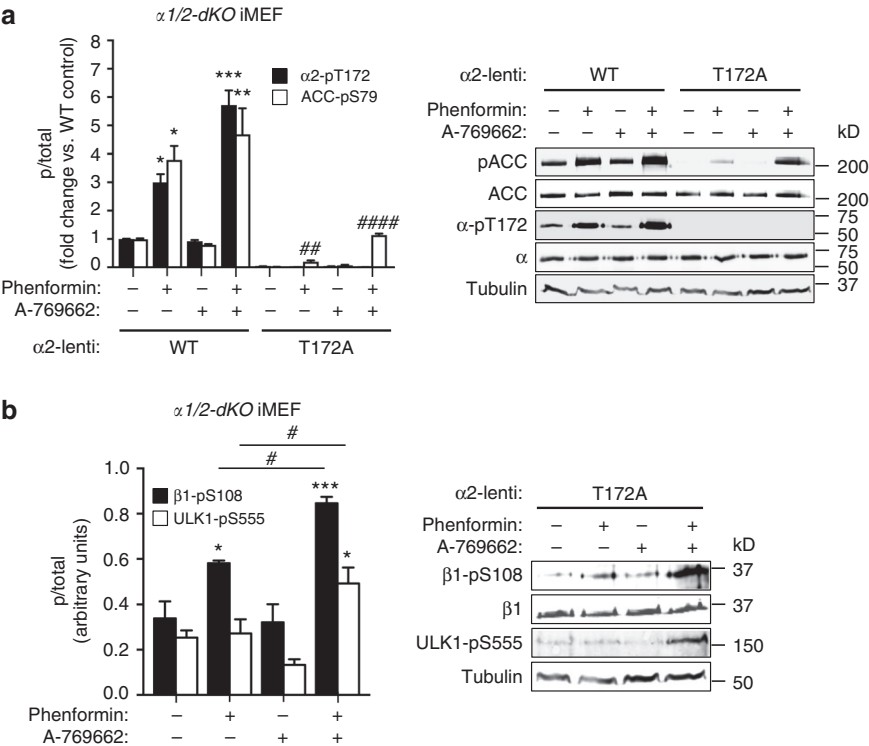

**Fig. 8** Cellular AMPK signaling occurs independently of α1-pThr172. Statistical analyses were performed using one-way ANOVA with post hoc Dunnett's multiple comparison test. **a** Immunoblots for α2-pThr172 and pACC from α1/2-dKO iMEFs-expressing α2 WT or T172A mutant, stimulated with 2 mM phenformin and/or 100 μM A-769662 for 1 h. $n = 3$, representative immunoblots shown. *Error bars*, mean pThr172 and pACC (fold change vs. untreated WT α2-expressing cells) ± s.e.m. ***$P < 0.001$, **$P > 0.01$, *$P < 0.05$ indicate significant increase compared to basal WT α2 cells. ####$P > 0.0001$, ##$P < 0.01$ indicate significant increase compared to basal α2(T172A) cells. **b** Immunoblots for β1-pSer108 and ULK1-pSer555 from α1/2-dKO iMEFs-expressing α2 T172A mutant, stimulated with 2 mM phenformin and 100 μM A-769662 for 1 h. $n = 3$, representative immunoblots shown. *Error bars*, mean Ser108 and Ser555 phosphorylation (arbitrary units) ± s.e.m. ***$P < 0.001$, *$P < 0.05$ indicate significant increase compared to basal. #$P < 0.05$ indicates significant increase compared to phenformin treated

upstream kinases and protein targets located at intracellular membranes[3, 30, 31]. We examined the requirement for β-myristoylation in directing Ser108 phosphorylation during cellular metabolic stress. We found that KI-α1β1γ1 containing a myristoylation-deficient β1-G2A mutant (non-myr) was insensitive to phenformin- or glucose starvation-induced phosphorylation of Ser108 in HEK293T cells (Fig. 7d). We examined the effect of β-myristoylation on cell-free ULK1-phosphorylation of Ser108 using myristoylated and non-myristoylated forms of KI-α1β1γ1. In contrast to our observation in cells, β-myristoylation resulted in significant suppression of ULK1-mediated Ser108 phosphorylation, which was not relieved by addition of AMP (Fig. 7e).

**pThr172 is not absolutely required for AMPK signaling**. To assess whether AMPK cellular signaling can be triggered independently of Thr172 phosphorylation, we expressed FLAG-tagged α2 WT or T172A mutant in α1/2-dKO iMEFs at similar levels (Fig. 8a). The T172A mutant possesses negligible basal and AMP-stimulated activities, but importantly can be sensitized to ADaM site metabolites/drugs through Ser108 phosphorylation[13]. In α1/2-dKO iMEFs-expressing WT α2, phenformin, but not A-769662, induced robust increases in pThr172 and pACC from a high basal level. In cells expressing α2(T172A), phenformin and A-769662, either alone or in combination, failed to induce Thr172 phosphorylation as expected. Phenformin, but not A-769662, induced a small increase in pACC from an undetectable basal level. Phenformin/A-769662 co-incubation resulted in a further 5.8-fold increase in pACC compared to phenformin alone, producing >30% the pACC signal in WT α2-expressing

cells treated with phenformin (Fig. 8a). Neither of these effects on pACC were detected in α1/2-dKO iMEFs transduced with empty lentivirus (Supplementary Fig. 6), confirming that AMPK was the only kinase phosphorylating ACC-Ser79 under these conditions. A similar phosphorylation profile was seen for Ser108 in α2 (T172A) cells, with phenformin inducing a small increase that was amplified in the additional presence of A-769662 (Fig. 8b). Phenformin/A-769662 co-incubation was the only condition that induced a detectable increase in ULK1-Ser555 phosphorylation, an AMPK substrate that is important for ULK1 activation[32] (Fig. 8b).

## Discussion

In this study, we demonstrate that β1-Ser108 in AMPK, a central co-ordinator of energy homeostasis, is a phosphorylation target for ULK1, a major regulator of autophagy initiation. Consequently, ULK1 sensitizes AMPK to A-769662 and salicylate, the active metabolic break-down product of acetylsalicilic acid (aspirin), independently of Thr172 phosphorylation (Fig. 4d). Salicylate stimulates fat utilization and reduces plasma fatty acids in vivo[15], reduces de novo lipogenesis in human hepatocytes and MEFs[33, 34], and reduces fatty acid and sterol synthesis in macrophages[35]. Daily aspirin prophylaxis is also associated with oncosuppression, an effect mirrored by the indirect AMPK activator metformin. Other β1-AMPK-specific direct activators MT63-78[36] and the indole acid derivative PF-06409577[37] have shown promise as treatments for either prostate cancer or diabetic neuropathy, respectively. Although not investigated, Ser108 phosphorylation is likely a requirement for

AMPK activation by these compounds; MT63-78 shares common structural features with A-769662, whereas the crystal structure of PF-06409577 bound to the ADaM site contains the phosphomimetic residue Asp instead of Ser108. Our finding that ULK1 phosphorylates Ser108 is significant given the emergence of this modification as a vital mechanistic constant for AMPK drugs. Discovery of a mechanism that induces AMPK drug sensitization independently of autophosphorylation also provides a potential strategy to treat non-small-cell lung and cervical carcinomas, associated with genetic loss of LKB1[38].

Accumulated evidence now depicts the ADaM site as an "orphan" allosteric site for an unidentified endogenous AMPK ligand: (i) β1-Ser108 and contacting α-residues are highly conserved among eukaryotes; (ii) the β1-ADaM site is transient and regulated; (iii) both ULK1 phosphorylation of Ser108, and activation of AMPK by small molecule drugs, share a common specificity for the β1-isoform; (iv) synergistic activation of unphosphorylated AMPK, orchestrated across all three AMPK subunits and topographically distant β1-ADaM and γ-sites[13], appears a highly intrinsic activation mechanism. Additionally, independent studies have described a disconnect between AMPK signaling (using elevated pACC as an index) and apparent AMPK activation (no increase in pThr172), most commonly in examinations of the role of skeletal muscle AMPK during exercise[39–42], but also in response to reactive oxygen species (HeLa cells)[43] and berberine (LAMTOR1-KO MEFs)[44]. We now demonstrate that significant cellular AMPK signaling can be triggered independently of pThr172 (Fig. 8a), although we are unable to determine whether this arises exclusively by Ser108 phosphorylation, AMP/drug synergistic activation of unphosphorylated enzyme, or a combination of both.

Applying quantitative phosphoproteomics to investigate a physiological role for Ser108 phosphorylation, we identified significant differences in the phosphorylation profiles of several cell cycle-associated proteins in response to the phosphomimetic mutant β1-S108E (Fig. 2). Our approach was limited in that we do not know whether potential metabolite(s) acting at the S108E-stabilized ADaM site were elevated in response to phenformin, thus our findings are likely an underestimation of the global effects of Ser108 phosphorylation. It should also be emphasized that the differences reflect both primary and downstream phosphorylation events, so that the phosphosites detected encompass more than direct AMPK sites. None of these phosphorylated residues have been identified as direct AMPK substrates, however, both DOCK7_Ser452 and NHRF1_Ser285 are potential candidates based on consensus with the AMPK substrate recognition motif (Supplementary Table 1). β1-S108E expression induced a significant increase in downstream phosphorylation of the PAK2 regulatory domain residue Ser141, an event required for full kinase activity and subsequent induction of the cytostatic and anti-apoptotic functions of full-length PAK2[45, 46]. Other downstream targets are associated with regulation of cell cycle arrest (Bin1; GO:0071156), p53-mediated cell cycle arrest (TB182; GO:0006977) and cell division (CND1; GO:0051301). Although functional roles for the majority of sites detected in our analysis are unknown, our findings with PAK2 in particular point to a role for β1-Ser108 phosphorylation in promoting energy stress-induced pro-survival pathways over cell death pathways. This model is consistent with other autophagy-inducing roles for AMPK, and Ser108-phosphorylation as a direct mechanism to activate AMPK by the autophagy initiator ULK1. Future studies will delineate the cellular signaling mechanisms specific to β1-Ser108 phosphorylation, and bridge the gap between AMPK and the other identified downstream substrates.

Closer examination of our results from MEFs expressing only α2(T172A) AMPK provides further insight into ULK regulation

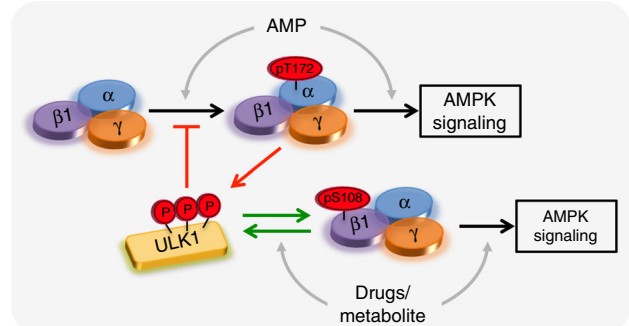

**Fig. 9** An integrated model for ULK1 regulation of β1-AMPK signaling. The initial ULK stimulus (e.g., Ser555 phosphorylation) is provided by AMPK, activated itself in response to energy stress and elevated AMP:ATP. Once activated, ULK1 suppresses Thr172 phosphorylation and AMPK sensitivity to AMP via negative feedback (red arrows)[20]. ULK1 simultaneously phosphorylates Ser108, sensitizing AMPK to drugs/metabolites acting at the ADaM site. This would promote AMP-independent AMPK signaling and maintain ULK1 activity via positive feedback (green arrows). In our cell model, activity of α2(T172A) AMPK is not elevated in response to AMP and requires additional stimulation with A-769662 to achieve ULK1 phosphorylation

of β1-AMPK signaling (Fig. 8). An integrated model, involving positive and negative feedback loops, is shown in Fig. 9. AMPK signaling in these cells is hampered by insensitivity to AMP; consequently phenformin is unable to stimulate ULK1 activity, contributing to the weak pSer108 response (Fig. 8b). Further addition of A-769662 was sufficient to rescue AMPK activity and stimulate phosphorylation of ULK1-Ser555 and Ser108, potentially via positive feedback. cis-autophosphorylation of Ser108 in the α2(T172A) mutant is unlikely since activity of the unphosphorylated complex can only be achieved synergistically; in the AMP/drug-bound state Ser108 is presumably sequestered to the ADaM site and away from the AMPK active site. The close correlation between pACC, pSer108 and ULK-pSer555 profiles across all conditions provides support for a positive feedback mechanism between ULK and AMPK (Fig. 8a, b). Consistent with this, co-incubation of cells expressing only KI-AMPK with phenformin/A-769662 did not induce a further increase in pSer108, compared to phenformin treatment alone (Fig. 3d). Our model raises the intriguing prospect that, on ULK1-populated membranes (autophagosome, mitochondria), AMPK may be desensitized to AMP through suppression of α1-Thr172 phosphorylation, yet simultaneously sensitized to an alternate regulatory ligand through phosphorylation of β1-Ser108.

Initial attempts to examine trans-phosphorylation of Ser108 in cells were confounded by strong background autophosphorylation signals in HEK293T endogenous AMPK, and high basal levels of pSer108 in immortalized MEFs (Fig. 6c). The cause of elevated pSer108 in iMEFs is unknown. Since KI-AMPK was not phosphorylated on Ser108 under basal conditions (Fig. 3d) we expect loss of a pSer108 phosphatase activity, leading to accumulation of Ser108 autophosphorylation in endogenous AMPK, may be a contributing factor. We were able to exploit the intramolecular "limitation" of Ser108 cis-autophosphorylation by using KI-AMPK in cells to uncouple background autophosphorylation from input by alternate signaling pathways. We identified conditions (glucose starvation, $H_2O_2$ and phenformin) under which Ser108 is phosphorylated independently of AMPK activity (Fig. 3b). These conditions also increase cellular LC3-II or ULK1-pSer555, both standard markers for autophagic activity[47–49]. Confirmation of ULK1-mediated phosphorylation of Ser108 in cells was provided by small molecule (Fig. 6a, b) or

genetic (Fig. 6d) inhibition of ULK activity, each of which resulted in almost complete loss of stress-induced Ser108 trans-phosphorylation.

ULK1 phosphorylation of Ser108 is specific to the AMPK β1-isoform (Fig. 5); this is likely due to sequence heterogeneity proximal to Ser108 in the β2-isoform that diverges from the ULK consensus found in the corresponding β1 residues[24]. Interestingly, β1 residues C-terminal to Ser108 (His109-Phe112) are considered unfavorable for ULK substrates. However, Ser108 is located at the end of a β-hairpin loop structure[12] and C-terminal residues may not participate in substrate recognition by ULK1. β1-Ser108 does not appear to be a favorable substrate for ULK2 (Fig. 4b, c), although we cannot rule out the possibility that the two ULK2 preparations used in this study may have possessed inherently low activity, given the current lack of knowledge regarding cellular regulation and bona fide substrates for this kinase. Overall, our findings are in agreement with autophagy network mapping showing interaction with AMPK β1 is restricted to ULK1, whereas ULK2 association is limited to AMPK β2[41]. Further investigation is warranted to determine subsets of ULK1 and ULK2 downstream targets in vivo.

Our results here, and those of others, provide strong evidence for potentiation of Ser108 phosphorylation by an AMP-myristoyl switch mechanism, in which AMP-induced conformational changes lead to ejection of the β-subunit myristoyl group from an intramolecular-binding site[3]. The exposed myristoyl group promotes AMPK targeting to intracellular membranes and, in this case, ULK1 co-localization. Cell treatments that promote pSer108 independently of autophosphorylation (glucose starvation, $H_2O_2$ and phenformin) (Fig. 3b) are associated with increased AMP/ATP ratio, either through mitochondrial toxicity or disruption of ATP production[50]. Treatments that did not induce Ser108 phosphorylation activate AMPK independently of AMP (A-769662, AICAR, ionomycin, amino-acid deprivation), or activate ULK independently of changes in adenylate charge (mTOR inhibitors) (Fig. 7a, c). The AMPK myristoyl-switching effect of ZMP (the metabolized product of AICAR and an AMP mimetic) has not yet been examined. Additionally, removal of either the AMP-sensing ability of AMPK with γ1(D245A) mutation (Fig. 7b), or β-subunit myristoylation with β1(G2A) mutation (Fig. 7d), abrogated glucose starvation- and/or phenformin-induced Ser108 phosphorylation. β1(G2A) mutation was previously shown to suppress AMPK partitioning to intracellular membranes following glucose starvation[3]. ULK1 is also recruited to membrane structures during autophagy initiation, notably the autophagosome formation sites located near the ER[51], and mitochondria[52]. ULK activity is associated with recruitment of other autophagy-associated proteins to the developing phagophore, including VPS34 and Beclin-1, both of which are phosphorylated by AMPK to achieve differential regulation of pro- and nonautophagy pathways[53]. Finally, β-myristoylation is strongly implicated in AMPK recruitment to LC3-containing puncta in the LKB1-deficient H23 human cancer cell line[31]. Collectively, these studies point to a close association of AMPK with ULK1 and the developing phagophore, and an important role for the exposed AMPK myristoyl group (myr switch ON) in targeting AMPK to phagophore membranes. A similar mechanism has been proposed to control AXIN-mediated AMPK recruitment to the late endosome/lysosome membrane for Thr172 phosphorylation and AMPK activation by LKB1[30]. An unexpected finding was myristoyl-dependent suppression of Ser108 phosphorylation by ULK1 in cell-free assays (Fig. 7e), indicating that AMPK adopts a conformation in the myristoyl-buried state (myr switch OFF) that is unfavorable for ULK1 phosphorylation of Ser108. Repression of Ser108 phosphorylation was not alleviated by AMP; either this demonstrates a

requirement for myristoyl-group membrane embedment, or an additional cellular component is required to derepress ULK phosphorylation through myristoyl group sequestration. These results may represent a protective mechanism to ensure Ser108 phosphorylation by ULK1 occurs exclusively at membrane surfaces.

In summary, we have identified an additional layer of communication between AMPK and ULK1, two major regulators of cellular energy homeostasis. ULK1 phosphorylation of Ser108, a post-translational modification important for AMPK drug action, may yield strategic opportunity to increase potency of AMPK-targeting therapeutics. Differential phosphorylation of AMPK at distinct cellular organelles/membranes raises the intriguing possibility of localized AMPK ligand sensitization. Myristoyl-switching is a consistent driving force for both mechanisms; whether secondary targeting signals are required to partition distinct AMPK complexes between organelles represents an area of great interest for the future.

## Methods

**Reagents.** DNA oligos were from Sigma (Supplementary Table 3). Antibodies for pan AMPK α (#2793, clone F6, 1:1000 dilution), FLAG (#2368, 1:1000 dilution), myc (#2276, clone 9B11, 1:1000 dilution), HA (#2367, clone 6E2, 1:1000 dilution), PAK1/2/3 (#2604, 1:1000 dilution), ULK1 (#4773, clone R600, 1:1000 dilution) and tubulin (#3873, clone DM1A, 1:1000 dilution), and phosphospecific antibodies for AMPK α-pThr172 (#2535, clone 40H9, 1:1000 dilution), AMPK β1-pSer108 (#4181, 1:1000 dilution), ACC-pSer79 (#3661, 1:1000 dilution), PAK1/2-pSer144/Ser141 (#2606, 1:1000 dilution), ULK1-pSer555 (#5869, clone D1H4, 1:1000 dilution), and ULK1-pSer757 (#6888, 1:1000 dilution) were from Cell Signaling Technology. AMPK-β1 antibody (#ab58175, 1:1000 dilution), A-769662 (#ab120335) and AICAR were from Abcam. IRDye 680RD- or 800CW-labeled anti-immunoglobulin G antibodies (1:10,000 dilution) and IRDye 680RD-labeled streptavidin (1:20,000 dilution) were from LI-COR Biosciences. Glutathione Sepharose 4B and Streptavidin Sepharose high performance were from GE Life Sciences. FLAG synthetic peptide (DYKDDDK) was provided by GL Biochem (Shanghai). Other synthetic peptides were from Purar Chemicals. All peptides were purified by reversed-phase chromatography and stored as lyophilized powder. ULK2 recombinant protein was from Abcam. SBI-0206965, AZD8055 and INK128 were from ApexBio. FuGENE HD transfection reagent was from Promega Corporation. All other reagents were from Sigma.

**Cell culture.** COS7 and HEK293T cell lines were purchased from American Type Culture Collection. All cell lines were maintained in Dulbecco's modified Essential medium (DMEM) containing 10% fetal bovine serum and antibiotics (penicillin, streptomycin) at 37 °C with 5% $CO_2$. To generate iMEF cell lines, MEFs were extracted from WT or homozygous AMPK β1β2 null embryos (days 12–14 post-coitum), generated by crossing homozygous β1 and β2 null mice[54]. WT and AMPK β1/2 double knockout (β1/2-dKO) MEFs were immortalized by Fugene HD-mediated transfection with an SV40 large-T antigen expression construct. AMPK α1/2 double knockout (α1/2-dKO) and ULK1/2 double knockout (ulk1/2-dKO) iMEFs were described previously[23, 29].

**Protein expression constructs.** All mutants were generated using QuikChange site-directed mutagenesis kits (Stratagene). All constructs were sequence verified. Mammalian cell expression constructs were gifts from Reuben Shaw (pcDNA3 mouse FLAG-ULK1 (Addgene #27636) and pcDNA3 mouse FLAG-ULK2 (Addgene #27637)), Noboru Mizushima (pME18s-3xHA-human FIP200 (Addgene #24303)) and Do-Hyung Kim (pRK5 human myc-Atg13 (Addgene #31965)). SV40 large-T antigen expression construct pBSSVD2005 was a gift from David Ron (Addgene plasmid #21826). Complementary DNA (cDNAs) for human AMPK α1 or α2 were generated with N-terminal FLAG-tag and cloned into pcDNA3 using XhoI/EcoRI (AMPK FLAG-α1) or XhoI/HindIII (AMPK FLAG-α2) restriction sites. Lentivirus expression constructs LeGO-iG2, second generation viral packaging vector psPax2 and ecotropic envelope vector pHCMV-EcoEnv were gifts from Carl Walkley (St Vincent's Institute of Medical Research). cDNAs for human AMPK β1 (WT and mutants), α1 (WT and kinase inactive D141A mutant), and α2 (WT and T172A mutant) were generated with C-terminal FLAG-tag and cloned into LeGO-iG2 using BamHI/NotI (β1) or EcoRI/NotI (α1 and α2) restriction sites.

**Protein expression and purification.** Heterotrimeric human AMPK (α1β1γ1 and α1β2γ1 expressed as N-terminal His$_6$-α fusions; WT or mutants as indicated) was expressed in E. coli strain Rosetta (DE3) and purified using nickel-Sepharose and size exclusion chromatography as described previously[13]. Heterotrimeric human AMPK (α1β1γ1 and α1β2γ1 expressed as an N-terminal GST-α fusion; WT or mutants as indicated) was expressed in HEK293T cells as described previously[3].

Briefly, HEK293T cells at 40% confluency were triply transfected with expression constructs for AMPK α-, β-, and γ-subunits, using transfection reagent FuGENE HD according to the manufacturer's protocols. Cells were harvested 48 h post-transfection in ice cold lysis buffer (50 mM Tris-HCl (pH 7.4), 150 mM NaCl, 50 mM NaF, 1 mM sodium pyrophosphate, 1 mM EDTA, 1 mM EGTA, 1% (v/v) Triton X-100, protease inhibitors). Lysates were clarified by centrifugation at 14,000 rpm for 5 min and flash frozen in liquid $N_2$ until processing. AMPK was purified on glutathione Sepharose 4B and eluted in 50 mM Tris-HCl, (pH 7.4), 100 mM NaCl, 10% glycerol supplemented with 20 mM glutathione. ULK1/2 and ULK1/FIP200/Atg13 complex was expressed by FuGENE HD transient transfection of HEK293T cells. Lysates were harvested after 48 h, clarified by centrifugation and ULK purified on anti-FLAG M2 affinity gel. ULK was eluted in 50 mM Tris-HCl (pH 7.4), 100 mM NaCl, 10% glycerol, 0.01% Tween-20 supplemented with 1 mg/ml FLAG peptide. CaMKK2 isoform 1 was produced in Sf21 insect cells as a C-terminal FLAG fusion as described previously[3]. Briefly, Sf21 cells were infected at a multiplicity of infection of 10 and harvested 72 h post-infection. Cell lysates were prepared as for recombinant AMPK, and protein was purified on anti-FLAG M2 affinity gel. CaMKK2 was eluted in 50 mM Tris-HCl (pH 7.4), 100 mM NaCl, 10% glycerol, 0.01% Tween-20 supplemented with 1 mg/ml FLAG peptide.

**Lentiviral-mediated AMPK expression and purification.** Ecotropic lentivirus was generated by transient transfection of HEK293T cells using calcium phosphate. Briefly, 1 day prior to transfection, $2–2.5 \times 10^6$ HEK293T cells were seeded per 10 cm culture dish. LeGO-iG2, psPax2 and pHCMV-EcoEnv plasmids (10 μg, 6.3 μg and 3.8 μg per 10 cm culture dish, respectively) were mixed together with 2 M $CaCl_2$ solution (244 mM final concentration in 500 μl). The $DNA/CaCl_2$ solution was added drop-wise, with vortexing, into 500 μl of 2xHEPES-buffered saline, pH 7.06. After 20 min incubation at RT, the mixture was transferred drop-wise onto HEK293T cell culture and incubated. Within 16 h of transfection, cell were washed with phosphate-buffered saline (PBS) and replaced with 6 ml fresh media. The lentivirus-containing supernatant was harvested after 48 and 72 h post-transfection and stored at −80 °C.

iMEFs were transduced in 6-well plates by spinoculation. Briefly, 1 day prior to transduction, $0.5 \times 10^5$ cells per well in a 6-well plate were seeded in 2 ml media. 200 μl lentivirus supernatant in the presence of 8 μg/ml polybrene in 2 ml media was spun for 98 min at 25 °C, $1100 \times g$ (Heraeus Megafuge 2.0 R). Lentivirus-containing media was replaced with an equal volume of fresh media after 24 h. Seventy-two hours post-transduction, iMEFs were incubated with fresh media for 1 h and treated as indicated. Cells were harvested by washing with ice-cold PBS, followed by rapid lysis in situ using 100 μl ice-cold lysis buffer (50 mM Tris.HCl (pH 7.4), 150 mM NaCl, 50 mM NaF, 1 mM sodium pyrophosphate, 1 mM EDTA, 1 mM EGTA, 1 mM dithiothreitol, 1% (v/v) Triton X-100 and protease inhibitors), and cellular debris was removed by centrifugation.

FLAG-tagged AMPK α1 or α2, and acetyl-CoA carboxylase (ACC), were isolated from iMEF cell lysates using anti-FLAG M2 affinity gel or Streptavidin Sepharose high performance, respectively. Immobilized ACC was washed extensively with buffer A and eluted in 2×sodium dodecyl sulfate polyacrylamide gel electrophoresis (SDS-PAGE) loading sample buffer for immunoblotting.

**Kinase activity assays.** ULK and AMPK activities were determined by phosphorylation of a synthetic peptide (S108tide: KLPLTRSHNNFVARRR, corresponding to AMPK β1(102–114) with three additional C-terminal Arg residues to promote binding to P81 phosphocellulose paper) using 200 μM [γ-$^{32}$P]ATP and 5 mM $MgCl_2$ in a 25 μl reaction volume at 30 °C. Reactions were terminated after 10 min by spotting 15 μl onto P81 phosphocellulose paper (Whatman) and washing in 1% phosphoric acid. Radioactivity was quantified by scintillation counting.

**Kinase screen.** Kinase screening (IKPT service) was performed by Kinexus, Canada. S108tide (100 μM) was screened against a panel of 284 selected Ser/Thr kinases with single replicate. Most assays were performed for 15 min duration, 50 μM [γ-$^{33}$P]ATP, in a 25 μl reaction volume at 30 °C.

**Generation of AMPK β2-pSer108 phosphospecific antibody.** A phosphorylated synthetic peptide (CSTKIPLIKpSHNDFVAILD, corresponding to AMPK β2 (100–117)) was coupled to keyhole limpet hemocyanin via the peptide N-terminal cysteine residue using the coupling reagent N-succinimidyl-3(-2-pyridyldithio) propionate. Rabbits were immunized with 2 mg of peptide conjugate initially in 50% (v/v) Freunds complete adjuvant and in 50% (v/v) Freunds incomplete adjuvant for subsequent immunizations. Rabbits were boosted fortnightly with 2 mg of peptide conjugate and bled 7 days after booster injections. The pSer108 antibody was then purified from serum by peptide affinity chromatography. Specificity for β2-pSer108 was evaluated by immunoblot against CaMKK2-phosphorylated AMPK α1β2γ1, and α1β1γ1.

**Immunoblotting.** Samples were electrophoresed by 12% SDS-PAGE and transferred to Immobilon FL polyvinylidine-flouride membrane (Millipore). The membrane was blocked with 2% nonfat dry milk in PBS +0.1% Tween 20 (PBS-T) for 1 h at room temperature. Membranes were incubated either overnight or for 1 h with 1° antibodies (diluted 1:500–1:2000 in PBS-T), prior to 30 min incubation with anti-rabbit or anti-mouse IgG 2° antibody fluorescently labeled with IR680 or IR800. Immunoreactive bands were visualized on an Odyssey membrane imaging system (LI-COR Biosciences) and densitometry analyses performed using integrated software. Uncropped western blot images associated with this study are presented in Supplementary Fig. 7.

**Quantitative global and phosphoproteomic analysis.** *β1/2-dKO* iMEFs, transduced with AMPK β1-S108A or β1-S108E, as described above, were incubated with 2 mM phenformin for 1 h. Treated lysates were acetone (−20 °C) precipitated and resuspended in 8 M urea/50 mM triethyl ammonium bicarbonate (TEAB) solution. Protein lysates were reduced with 10 mM tris(2-carboxyethyl)phosphine (TCEP) for 45 min at 37 °C, and alkylated with 55 mM iodoacetamide for 30 min at room temperature in the dark. Samples were diluted to 1 M urea in 25 mM TEAB and trypsin (1:60, w/w) digested overnight at 37 °C. Digested tryptic peptides were cleaned up using Oasis HLB (hydrophilic lipophilic balance) solid phase extraction (SPE) cartridges (Waters) and freeze-dried overnight. Resuspended tryptic digests were used for stable-isotope dimethyl labeling as described previously[55]. β1-S108A and β1-S108E derived tryptic peptides were used for light and medium labeling, respectively. Labeled peptides were mixed (1:1) and an aliquot (15 μl) was taken out and run on liquid chromatography–mass spectrometry (LC-MS)/MS for total proteome changes. Remaining mixed labeled peptides were SPE cleaned up and freeze-dried before used for $TiO_2$ phosphopeptides enrichment. Phosphopeptide enrichment by $TiO_2$ microcolumns was carried out as described previously[56]. Mixed labeled peptides and $TiO_2$ enriched phosphopeptides were analyzed by LC-MS/MS using a Q-Exactive plus mass spectrometer (Thermo Scientific) fitted with nanoflow reversed-phase-high-performance liquid chromatography (HPLC) (Ultimate 3000 RSLC, Dionex). The nano-LC system was equipped with an Acclaim Pepmap nano-trap column (Dionex—C18, 100 Å, 75 μm × 2 cm) and an Acclaim Pepmap RSLC analytical column (Dionex—C18, 100 Å, 75 μm × 50 cm). Typically for each LC-MS/MS experiment, 5 μl of the peptide mix was loaded onto the enrichment (trap) column at an isocratic flow of 5 μl/min of 3% $CH_3CN$ containing 0.1% formic acid for 6 min before the enrichment column is switched in-line with the analytical column. The eluents used for the LC were 0.1% v/v formic acid (solvent A) and 100% $CH_3CN$/0.1% formic acid v/v (solvent B). The gradient used was 3% B to 25% B for 23 min, 25% B to 40% B in 2 min, 40% B to 80% B in 2 min and maintained at 85% B for the final 2 min before equilibration for 9 min at 3% B prior to the next analysis. All spectra were acquired in positive mode with full scan MS spectra scanning from $m/z$ 375–1400 at 70,000 resolution with AGC target of 3e6 with maximum accumulation time of 50 ms. Lockmass of 445.120024 was used. The 15 most intense peptide ions with charge states ≥2–5 were isolated with isolation window of 1.2 $m/z$ and fragmented with normalized collision energy of 30 at 35,000 resolution with AGC target of 1e5 with maximum accumulation time of 120 ms. Underfill threshold was set to 2% for triggering of precursor for MS2. Dynamic exclusion was activated for 30 s. Mass spectrometric raw data were processed and analyzed using Proteome Discoverer 2.1 (Thermo Scientific) with Mascot search algorithm against mouse SwissProt database. Perseus 1.5.6.0 was used for further data analysis[57]. IPA software (QIAGEN Redwood City) was used for network and pathway analysis.

**In-gel digestion of protein bands and mass spectrometry.** SDS-PAGE protein bands were excised and simultaneously reduced and alkylated with 10 mM TCEP and 40 mM 2-choloracetamide for 1 h at room temperature. Proteins were digested with 13 ng/μl of sequencing grade trypsin (Promega) overnight at 37 °C and peptides desalted using in-house made microC18 columns (3 M empore). Peptides were resuspended in 0.1% formic acid, 2% acetonitrile and analyzed on a Dionex 3500RS nanoUHPLC coupled to an Orbitrap Fusion mass spectrometer with Tune v2.0.1258 in positive mode. Peptides were separated using an in-house packed 75 μm × 40 cm pulled column (1.9 μm particle size, C18AQ; Dr Maisch, Germany) with a gradient of 2–30% acetonitrile containing 0.1% FA over 60 min at 250 nl/min at 55 °C. An MS1 scan was acquired from 350–1550 (120,000 resolution, 5e5 AGC, 100 ms injection time) followed by MS/MS data-dependent acquisition with HCD and detection in the Orbitrap (60,000 resolution, 2e5 AGC, 120 ms injection time, 40 NCE, 2.0 $m/z$ quadrupole isolation width) and, EThcD and detection in the orbitrap (60,000 resolution, 2e5 AGC, 120 ms injection time, calibrated charge-dependent ETD reaction times [2 + 121; 3 + 54; 4 + 30; 5 + 20; 6 + 13; 7 + ; 10 ms], 25 NCE for HCD supplemental activation, 2.0 $m/z$ quadrupole isolation width). All raw data were analyzed with MaxQuant v1.5.3.25[58] and searched against the human UniProt database with default settings including phosphorylation of S, T, and Y as a variable modification and match between runs enabled.

**Nucleotide measurements.** Adenine nucleotides from HEK293T perchlorate extracts were measured by LC-MS on an ABISCIEX 5500 QTRAP mass spectrometer[16]. AEC was calculated from ratios of [AMP], [ADP], and [ATP] (Equation 1):

$$AEC = \frac{[ATP] + (0.5[ADP])}{[ATP] + [ADP] + [AMP]} \quad (1)$$

**Statistical analysis**. The data are presented as mean values ± s.e.m. of at least three independent experiments. The unpaired two-tailed Student's *t*-test was used for all comparisons unless stated.

**Data availability**. The mass spectrometry proteomics data have been deposited to the ProteomeXchange Consortium via the PRIDE[59] partner repository with the data set identifier PXD006365. The authors declare that all other data supporting the findings of this study are available within the paper and its Supplementary Information files, or are available from the authors upon reasonable request.

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

## Acknowledgements

This work was supported by grants from the Australian Research Council (ARC) and the National Health and Medical Research Council (NHMRC). G.R.S. is a Canada Research Chair in Metabolism and Obesity and the J. Bruce Duncan Chair in Metabolic Diseases. B.E.K. is an NHMRC Research Fellow. J.S.O. is an ARC Future Fellow. Supported in part by the Victorian Government's Operational Infrastructure Support Program.

## Author contributions

T.A.D., N.X.Y.L., and J.S.O. designed and coordinated the study. T.A.D., N.X.Y.L., J.W.S., A.H., B.L.P., K.R.W.N., M.T.O., and C.G.L. performed the experiments. S.G., M.K., and B.V. provided immortalized mouse embryonic fibroblasts. J.W.S., G.R.S., K.S., and B.E.K. provided intellectual input. All authors contributed to writing the manuscript.

## Additional information

**Competing interests:** The authors declare no competing financial interests.

