## [Peer Review File · Nature Communications]

Reviewers' comments:

Reviewer #1 (Remarks to the Author):

The manuscript by Dite and colleagues reports on the phosphorylation of S108 in PRKAB1 by ULK1, and the potential role that this might have in AMPK regulation. The strength of the study is that it is novel and provides another potential link between two important pathways. The weakness of the study is that some critical experiments are missing, the experimental design involves many different systems that are in some cases convoluted, and some of the data is difficult to interpret. Moreover, enthusiasm for the study is limited by the lack of an obvious physiological role for the system described.

Major concerns

1. An issue that the authors have not fully addressed is whether trans phosphorylation of S108 (e.g. in the experiment shown in Fig. 2) could be due to AMPK (endogenous AMPK present in the cells). In order to rule this out, the kinase inactive form of AMPK should be expressed in cells lacking endogenous AMPK activity (ideally, PRKAA1/PRKAA2 double KO cells). In this system, any S108 phosphorylation would have to be due to another kinase (not AMPK). In my view, this is an absolutely essential experiment, since as it stands an alternative explanation to the authors' hypothesis would be that AMPK (endogenous activity present in the cells) could account for the S108 phosphorylation observed (and see additional comments below).
2. It is not clear to me how the quantification of the T172 and S108 phosphorylation can be compared to each other (e.g. in Fig. 2b). The fold increase is presumably dependent on the "basal" level of phosphorylation – but what does this mean e.g. in terms of stoichiometry? What would represent 0 and 100% phosphorylation in each case? So, in the example shown in Fig. 2b, T172 phosphorylation is relatively high in the basal state (why?) whereas S108 is low. In this case, the changes in T172 are modest (because the basal level is high – in fact, T172 phosphorylation actually appears to decrease in the cells treated with AICAR and A769662, although it's not clear what this might be due to). Changes in S108 appear much greater, but this is skewed because of the low signal in the basal state. Some additional way of quantification seems to be needed in order to interpret these data.
3. What is the stoichiometry of S108 phosphorylation by ULK1 in vitro (Fig. 3)? In the text (page 7) it is stated that ULK1 was >35% activity relative to AMPK. Does this mean that AMPK phosphorylates the S108 peptide much better than ULK1 (again supporting the idea that AMPK itself may be the bona fide S108 kinase)?
4. The authors introduce a new mutant (C176S). Some description of the effect of this mutant on AMPK activity (including allosteric regulation by AMP, A769662) would seem appropriate and necessary here, since it has not been described previously.
5. The results shown in Fig. 5d provide very good evidence that ULK1/2 are not essential for S108 phosphorylation (presumably because of AMPK autophosphorylation). This raises the key issue of what (if any) the role for ULK1 S108 phosphorylation might be. The authors invoke an interesting experimental scenario to try and explain this by expressing kinase inactive AMPK in the Ulk1/2 KO MEFs. Even so, there is still some S108 phosphorylation in the double Ulk1/2 KO MEFs in the presence of phenformin (Fig. 5d). A major problem, therefore, is under what conditions might ULK1 play a role in regulating AMPK if S108 can be phosphorylated efficiently in its absence? Inclusion of a cellular system (or in vivo model) where this could be demonstrated would add greatly to the current findings and broaden the appeal of the study.

Reviewer #2 (Remarks to the Author):

In the manuscript by Dite et al. the authors reveal a crosstalk between ULK1 and AMPK. They propose that ULK1 phosphorylates Ser108 in the AMPK beta1 regulatory subunit and sensitizes AMPK to allosteric drugs binding to the ADaM site. These findings are important as they suggest a mechanism by which the efficacy of AMPK activators could be modulated by changes in Ulk1 signaling. In addition, the authors also provide substantial evidence that Ulk1-mediated phosphorylation is dependent on AMPK beta1 myristoylation associated with elevated AMP/ATP ratio. This study is well executed and will be of interest to the community.

Specific comments

- 1) The observation that certain stress conditions can promote Ser108 phosphorylation in trans is intriguing. For example, in glucose starvation condition, it is unclear to what extent AMPK beta1 Ser108 phosphorylation could be dependent on Ulk1. Previous studies have shown that glucose starvation induces AMPK recruitment to the late endosome/ lysosome membrane. It would be of great interest to determine the contribution of AMPK recruitment at the autophagosome or lysosome membrane structures.
- 2) The authors do not mention the other kinases that phosphorylate S108. In other words, is it possible that in addition to ULK1, other relevant kinases phosphorylate and participate to the activation of AMPK in particular settings?
- 3) Metformin is commonly used to treat type 2 diabetes and there is increasing interest in understanding its physiological effects because of its potential to fight cancer. The authors only reported results from phenformin treated cells. Is metformin mimicking the effect of phenformin on Ser108 phosphorylation? What is the dependence of metformin-mediated activation on AMPK beta Ser108 phosphorylation and myristoylation? A recent study has reported that metformin-induced AMPK activation is dependent on the assembly of the AXIN-LKB1-AMPK complex.
- 4) The authors clearly demonstrate the ability of phenformin or glucose starvation to induce Ser108 phosphorylation but there is no clear view on the importance of this post-translational modification on their efficacy to activate overall AMPK signaling. Does impaired Ser108 phosphorylation (e.g. in ulk-1/2-dKO cells or ulk-1/2-dKO cells expressing S108A mutant) correlate with a decrease in the phosphorylation of AMPK targets?
- 5) The authors demonstrate that ULK1 sensitizes AMPK to allosteric drugs binding the AdAM site through a mechanism dependent on an AMP-induced myristoyl switch mechanism. Cell treatments identified in this study to promote Ser108 phosphorylation are glucose starvation, H₂O₂ incubation and phenformin treatment. They claim that these findings are important for improving the efficacy of AMPK activating drugs. However, how this AMP-induced myristoyl switch would be induced to enhance allosteric drugs efficacy in patients is not discussed at all.

Reviewer #3 (Remarks to the Author):

This manuscript presents a well-performed, rigorous examination by Oakhill and colleagues of regulation of AMPK activity via modulation of beta1 Ser108 phosphorylation, previously shown to be critical for the binding and activation of AMPK by A769662, salicylate, and related small molecule activators. The surprise finding here is that the autophagy kinase ULK1, a well established substrate of AMPK, may be mediating a feedback - transphosphorylation event on AMPK in a beta1 specific fashion. This study is well-performed and tackles this complex set of regulatory events in a well-depicted fashion critical for most readers to understand the importance and functional connections. I should add that the Discussion here is long, but I think the authors raise

a number of key issues that would serve readers best if its length was not reduced greatly.

Some surprises arise in their findings, including that ULK1 which is activated by mTOR catalytic inhibitors is not sufficient to phosphorylate beta1 Ser108 in vivo. This and many other pieces of data do support the authors models of subcellular pools of AMPK and ULK1 which may mediate a specific subset of AMPK actions.

There is much here in a very advanced and well-analyzed fashion such that this reviewer believes the manuscript to be appropriate for publication in Nat Comm with minimal revision. In fact only one significant experimental question that might significantly enhance the findings.

Major point 1.

One major question that arises in this reviewers mind which should not be experimentally difficult for the authors to address. The question is how much does phosphorylation of beta1 Ser108 by AMPK α versus ULK1 play in cells in terms of overall AMPK activation and downstream target engagement in response to any particular stimuli. This could be easily examined in WT vs ULK1/2 DKO MEFs, treated with the panel of stimuli in Figure 2B, but immunoblotting the WT vs ULK1/2 DKO MEF lysates for P-ACC, P-raptor. Is ULK1 needed for maximal induction of P-ACC after some stimuli? Does P-raptor behave differently? One might imagine a common pool of mTORC1 and ULK1 with AMPK at lysosomes or other subcellular locations that might be most sensitive to the impact of the authors myrisoltyation model. If so, perhaps P-raptor would be affected in the ULK1 DKO cells whereas ACC would not be affected by ULK1/2 deficiency.

Reviewer #1:

We thank the Reviewer for his/her constructive comments. We have performed additional experiments where appropriate and addressed concerns or suggestions individually below:

1. An issue that the authors have not fully addressed is whether trans phosphorylation of S108 (e.g. in the experiment shown in Fig. 2) could be due to AMPK (endogenous AMPK present in the cells). In order to rule this out, the kinase inactive form of AMPK should be expressed in cells lacking endogenous AMPK activity (ideally, PRKAA1/PRKAA2 double KO cells). In this system, any S108 phosphorylation would have to be due to another kinase (not AMPK). In my view, this is an absolutely essential experiment, since as it stands an alternative explanation to the authors' hypothesis would be that AMPK (endogenous activity present in the cells) could account for the S108 phosphorylation observed (and see additional comments below).

We have performed the suggested experiment using AMPK α 1/2 dKO MEFs transduced to express kinase inactive α 1. This experiment, along with validation of the cell model, is detailed in new Figs 2c and d. Phenformin induced S108 phosphorylation in these cells, which was not increased with A-769662 co-incubation. This confirms that S108 is a substrate for an alternate kinase(s). While we cannot exclude S108 autophosphorylation in *trans*, all evidence presented (this study and previously (Scott *et al*, 2014)) is supportive of a *cis*-autophosphorylation mechanism.

2. It is not clear to me how the quantification of the T172 and S108 phosphorylation can be compared to each other (e.g. in Fig. 2b). The fold increase is presumably dependent on the "basal" level of phosphorylation – but what does this mean e.g. in terms of stoichiometry? What would represent 0 and 100% phosphorylation in each case? So, in the example shown in Fig. 2b, T172 phosphorylation is relatively high in the basal state (why?) whereas S108 is low. In this case, the changes in T172 are modest (because the basal level is high – in fact, T172 phosphorylation actually appears to decrease in the cells treated with AICAR and A769662, although it's not clear what this might be due to). Changes in S108 appear much greater, but this is skewed because of the low signal in the basal state. Some additional way of quantification seems to be needed in order to interpret these data.

The purpose of the experiment was to demonstrate whether, and under what conditions, S108 is phosphorylated in *trans*. In this respect the results are illuminating. We agree with the Reviewer that Fig 2b presentation is misleading due to low basal pS108 signal; we have now removed the pT172 blot and graph to improve clarity. The remaining pS108 blot more succinctly shows conditions under which S108 is *trans*-phosphorylated. The newly included Fig 2d confirms the existence of a S108 kinase besides AMPK. Including any estimate of stoichiometries for S108 phosphorylation in Fig 2b would be misleading since we used an over-expression system.

3. What is the stoichiometry of S108 phosphorylation by ULK1 in vitro (Fig. 3)? In

the text (page 7) it is stated that ULK1 was >35% activity relative to AMPK. Does this mean that AMPK phosphorylates the S108 peptide much better than ULK1 (again supporting the idea that AMPK itself may be the bona fide S108 kinase)?

We are not of the opinion that there is just one bona fide S108 kinase. α 1-Ser487, for instance, is an AMPK autophosphorylation site but also a well characterized substrate for multiple other kinases. Nor do we regard stoichiometry of phosphorylation using peptide/purified enzyme substrates as meaningful. Conditions for the peptide screen are proprietary and were most likely not optimized for individual kinases, so a direct comparison of specific activity is unwarranted. Nevertheless, we have found that S108tide demonstrates marginally better substrate affinity for AMPK (15 μ M) than ULK1 (76 μ M), but whether this is relevant to the situation *in vivo* is unclear.

4. The authors introduce a new mutant (C176S). Some description of the effect of this mutant on AMPK activity (including allosteric regulation by AMP, A769662) would seem appropriate and necessary here, since it has not been described previously.

The initial manuscript included data showing synergistic activation of unphosphorylated C176S by AMP/A-769662 or C2/A-769662 co-incubation (Supp Fig 2e), and AMP allosteric regulation of CaMKK2-treated C176S (Supp Fig 2f, now Supp Fig 2g). We have now added A-769662 activation of CaMKK2-activated C176S (new Supp Fig 2f). Combined, our data demonstrate no significant difference to WT in terms of small molecule allosteric regulation, and show that AMP, C2 and ADaM sites (and associated activating mechanisms) in the C176S mutant are fully preserved.

5. The results shown in Fig. 5d provide very good evidence that ULK1/2 are not essential for S108 phosphorylation (presumably because of AMPK autophosphorylation). This raises the key issue of what (if any) the role for ULK1 S108 phosphorylation might be. The authors invoke an interesting experimental scenario to try and explain this by expressing kinase inactive AMPK in the Ulk1/2 KO MEFs. Even so, there is still some S108 phosphorylation in the double Ulk1/2 KO MEFs in the presence of phenformin (Fig. 5d). A major problem, therefore, is under what conditions might ULK1 play a role in regulating AMPK if S108 can be phosphorylated efficiently in its absence? Inclusion of a cellular system (or in vivo model) where this could be demonstrated would add greatly to the current findings and broaden the appeal of the study.

With great respect we disagree with the Reviewer's interpretation of data presented in Fig5d; also we do not think that ULK1 represents the only mechanism for S108 phosphorylation. Fig 5d (lane 4, pS108 IB) shows almost complete loss of phenformin-induced pS108 on kinase inactive AMPK expressed in Ulk dKO, vs WT MEFs. The very low residual pS108 detected on KI-AMPK in phenformin treated Ulk dKO MEFs (that we assume the Reviewer is referring to) has most likely arisen from other S108 kinases not fully characterized in this study.

We have made extensive effort to devise a cell model to determine the contribution of ULK-mediated S108 phosphorylation to cellular AMPK signalling; we conclude this is not possible to achieve unequivocally with current tools.

i) Our interpretation is that ULK and AMPK form negative (down-regulation of pT172) and positive (S108 phosphorylation) regulatory feedback loops, preventing disruption of either signalling system in isolation.

ii) ULK phosphorylation of Ser108, and S108 autophosphorylation, are also intrinsically linked in their requirement for an initial AMP signal. Hence we are unable to use endogenous AMPK to delineate the signalling effects of the two mechanisms.

iii) We are prevented from using ulk1/2 dKO or WT immortalized MEFs due to abnormally high basal pS108 on endogenous AMPK in these cells. We discuss this may arise from loss of a pS108 phosphatase during transformation.

iv) We do not yet know the identity of specific substrates targeted by S108-phosphorylated AMPK (e.g. at the autophagosome) in contrast to T172-phosphorylated enzyme. Some of these may be entirely novel and will be the subject of future investigations.

v) We are reluctant to use the small compound ULK1 inhibitor 6965 due to possible off-target effects.

Our model for ULK regulation of AMPK proposes a scenario where AMPK signalling occurs independently of T172 phosphorylation. We have included new data using AMPK α 1/2 dKO iMEFs transduced with an α 2(T172A) mutant. This mutant is catalytically active and a suitable substrate for ULK1, but cannot be activated by upstream kinases LKB1/CaMKK2, nor allosterically by AMP alone. Validation of the cell model forms a new Fig 2c. The experiment forms new Figs 7a & b, Supp Fig 5, and extra discussion points (we have trimmed other discussion sections to accommodate). We detected low level pACC in response to phenformin, which was increased a further 3.5-fold with additional incubation with A-769662. Co-incubation also induced S108 and ULK1-S555 phosphorylation, indicative of positive feedback. Maximal pACC was ~45% compared to similarly treated cells expressing WT α 2. We did not detect pACC in cells expressing empty lentivirus in response to phenformin/A-769662 (Supp Fig 5), demonstrating that AMPK is the exclusive kinase for this target under these conditions. We cannot unambiguously attribute this AMPK activity as either pS108-mediated, due to AMP/drug synergy of unphosphorylated AMPK, or a combination of both, and state this clearly in the discussion. However, this experiment provides definitive proof that significant cellular AMPK signalling can occur independently of pT172, which will be of interest to the field. We are aware this is a complicated mechanism and have now included a cartoon of our model in Fig 8 to aid the reader.

Reviewer #2:

We thank the Reviewer for his/her constructive comments. We have performed additional experiments where appropriate and address concerns or suggestions individually below:

1. The observation that certain stress conditions can promote Ser108 phosphorylation in trans is intriguing. For example, in glucose starvation condition, it is unclear to what extent AMPK beta1 Ser108 phosphorylation could be dependent on Ulk1. Previous studies have shown that glucose starvation induces AMPK recruitment to the late endosome/ lysosome membrane. It would be of great interest to determine the contribution of AMPK recruitment at the autophagosome or lysosome membrane structures.

The Reviewer makes a good suggestion for future direction; specific recruitment to different cellular organelles eg mitochondria, lysosomes, autophagosomes most likely dictates many AMPK signalling functions. This will certainly be investigated but we do not consider these experiments fall within the scope of the current study.

2. The authors do not mention the other kinases that phosphorylate S108tide. In other words, is it possible that in addition to ULK1, other relevant kinases phosphorylate and participate to the activation of AMPK in particular settings?

We have listed some of the other candidate S108 kinases arising from our peptide screen (page 8). We agree that some of these kinases may bear significance in certain scenarios e.g. cell cycling in the context of NEKs. However, our experiments in Fig5b and d provide strong evidence that ULK1 is the dominant S108 kinase operating in *trans* during periods of elevated AMP.

3) Metformin is commonly used to treat type 2 diabetes and there is increasing interest in understanding its physiological effects because of its potential to fight cancer. The authors only reported results from phenformin treated cells. Is metformin mimicking the effect of phenformin on Ser108 phosphorylation? What is the dependence of metformin-mediated activation on AMPK beta Ser108 phosphorylation and myristoylation? A recent study has reported that metformin-induced AMPK activation is dependent on the assembly of the AXIN-LKB1-AMPK complex.

Both metformin and phenformin are biguanides that impair mitochondrial respiration through inhibition of complex I, leading to reduced ATP production and AMPK activation. Their metabolic effects are very similar (Janzer et al. 2014, PNAS 111; p10574). Phenformin is regarded as a more potent version of metformin, hence its use in the present study. We have not examined the effect of metformin on ULK-mediated S108 phosphorylation, nor dependence on myristoylation, but expect it would produce similar results at higher dose.

4) The authors clearly demonstrate the ability of phenformin or glucose starvation to induce Ser108 phosphorylation but there is no clear view on the importance of

this post-translational modification on their efficacy to activate overall AMPK signaling. Does impaired Ser108 phosphorylation (e.g. in ulk-1/2-dKO cells or ulk-1/2-dKO cells expressing S108A mutant) correlate with a decrease in the phosphorylation of AMPK targets?

We are grateful to the Reviewer for suggesting these experiments. We have made extensive effort to devise a cell model to determine the contribution of ULK-mediated S108 phosphorylation to cellular AMPK signalling; we conclude this is not possible to achieve unequivocally with current tools.

i) Our interpretation is that ULK and AMPK form negative (down-regulation of pT172) and positive (S108 phosphorylation) regulatory feedback loops, preventing disruption of either signalling system in isolation.

ii) ULK phosphorylation of Ser108, and S108 autophosphorylation, are also intrinsically linked in their requirement for an initial AMP signal. Hence we are unable to use endogenous AMPK to delineate the signalling effects of the two mechanisms.

iii) We are prevented from using ulk1/2 dKO or WT immortalized MEFs due to abnormally high basal pS108 on endogenous AMPK in these cells. We discuss this may arise from loss of a pS108 phosphatase during transformation.

iv) We do not yet know the identity of specific substrates targeted by S108-phosphorylated AMPK (e.g. at the autophagosome) in contrast to T172-phosphorylated enzyme. Some of these may be entirely novel and will be the subject of future investigations.

v) We are reluctant to use the small compound ULK1 inhibitor 6965 due to possible off-target effects.

Our model for ULK regulation of AMPK proposes a scenario where AMPK signalling occurs independently of T172 phosphorylation. We have included new data using AMPK $\alpha 1/2$ dKO iMEFs transduced with an $\alpha 2$ (T172A) mutant. This mutant is catalytically active and a suitable substrate for ULK1, but cannot be activated by upstream kinases LKB1/CaMKK2, nor allosterically by AMP alone. Validation of the cell model forms a new Fig 2c. The experiment forms new Figs 7a & b, Supp Fig 5, and extra discussion points (we have trimmed other discussion sections to accommodate). We detected low level pACC in response to phenformin, which was increased a further 3.5-fold with additional incubation with A-769662. Co-incubation also induced S108 and ULK1-S555 phosphorylation, indicative of positive feedback. Maximal pACC was ~45% compared to similarly treated cells expressing WT $\alpha 2$. We did not detect pACC in cells expressing empty lentivirus in response to phenformin/A-769662 (Supp Fig 5), demonstrating that AMPK is the exclusive kinase for this target under these conditions. We cannot unambiguously attribute this AMPK activity as either pS108-mediated, due to AMP/drug synergy of unphosphorylated AMPK, or a combination of both, and state this clearly in the discussion. However, this experiment provides definitive proof that significant cellular AMPK signalling can occur independently of pT172, which will be of interest to the field. We are aware this is a complicated mechanism and have now included a cartoon of our model in Fig 8 to aid the reader.

5) The authors demonstrates that ULK1 sensitizes AMPK to allosteric drugs binding the AdAM site through a mechanism dependent on an AMP-induced myristoyl

switch mechanism. Cell treatments identified in this study to promote Ser108 phosphorylation are glucose starvation, H2O2 incubation and phenformin treatment. They claim that these findings are important for improving the efficacy of AMPK activating drugs. However, how this AMP-induced myristoyl switch would be induced to enhance allosteric drugs efficacy in patients is not discussed at all.

We now briefly mention in the discussion the implications of this study for AMPK drug strategies.

Reviewer #3:

We thank the Reviewer for his/her constructive comments. We have performed additional experiments where appropriate and addressed concerns or suggestions individually below:

1. One major question that arises in this reviewers mind which should not be experimentally difficult for the authors to address. The question is how much does phosphorylation of beta1 Ser108 by AMPK α versus ULK1 play in cells in terms of overall AMPK activation and downstream target engagement in response to any particular stimuli. This could be easily examined in WT vs ULK1/2 DKO MEFs, treated with the panel of stimuli in Figure 2B, but immunoblotting the WT vs ULK1/2 DKO MEF lysates for P-ACC, P-raptor. Is ULK1 needed for maximal induction of P-ACC after some stimuli? Does P-raptor behave differently? One might imagine a common pool of mTORC1 and ULK1 with AMPK at lysosomes or other subcellular locations that might be most sensitive to the impact of the authors myrisoltyation model. If so, perhaps P-raptor would be affected in the ULK1 DKO cells whereas ACC would not be affected by ULK1/2 deficiency.

We are grateful to the Reviewer for suggesting these experiments. We have made extensive effort to devise a cell model to determine the contribution of ULK-mediated S108 phosphorylation to cellular AMPK signalling; we conclude this is not possible to achieve unequivocally with current tools.

- i) Our interpretation is that ULK and AMPK form negative (down-regulation of pT172) and positive (S108 phosphorylation) regulatory feedback loops, preventing disruption of either signalling system in isolation.
- ii) ULK phosphorylation of Ser108, and S108 autophosphorylation, are also intrinsically linked in their requirement for an initial AMP signal. Hence we are unable to use endogenous AMPK to delineate the signalling effects of the two mechanisms.
- iii) We are prevented from using ulk1/2 dKO or WT immortalized MEFs due to abnormally high basal pS108 on endogenous AMPK in these cells. We discuss this may arise from loss of a pS108 phosphatase during transformation.
- iv) We do not yet know the identity of specific substrates targeted by S108-phosphorylated AMPK (e.g. at the autophagosome) in contrast to T172-phosphorylated enzyme. Some of these may be entirely novel and will be the subject of future investigations.
- v) We are reluctant to use the small compound ULK1 inhibitor 6965 due to possible off-target effects.

Our model for ULK regulation of AMPK proposes a scenario where AMPK signalling occurs independently of T172 phosphorylation. We have included new data using AMPK α 1/2 dKO iMEFs transduced with an α 2(T172A) mutant. This mutant is catalytically active and a suitable substrate for ULK1, but cannot be activated by upstream kinases LKB1/CaMKK2, nor allosterically by AMP alone. Validation of the cell model forms a new Fig 2c. The experiment forms new Figs 7a & b, Supp Fig 5, and extra discussion points (we have trimmed other discussion sections to accommodate). We detected low level pACC in response to phenformin, which was increased a further 3.5-fold with additional incubation with A-769662. Co-incubation also induced S108 and ULK1-S555

phosphorylation, indicative of positive feedback. Maximal pACC was ~45% compared to similarly treated cells expressing WT $\alpha 2$. We did not detect pACC in cells expressing empty lentivirus in response to phenformin/A-769662 (Supp Fig 5), demonstrating that AMPK is the exclusive kinase for this target under these conditions. We cannot unambiguously attribute this AMPK activity as either pS108-mediated, due to AMP/drug synergy of unphosphorylated AMPK, or a combination of both, and state this clearly in the discussion. However, this experiment provides definitive proof that significant cellular AMPK signalling can occur independently of pT172, which will be of interest to the field. We are aware this is a complicated mechanism and have now included a cartoon of our model in Fig 8 to aid the reader.

Reviewers' comments:

Reviewer #1 (Remarks to the Author):

The authors have made a number of changes to the manuscript in order to respond to my initial comments. These have improved the manuscript, although I think it is still unclear what, if any, is the physiological significance of the findings. There are a couple of points arising in the revised manuscript that the authors should address.

1. In the new version of Figure 2d, a blot for the parental cells (or empty lenti virus control) should be included.
2. For Figure 3 (and referring back to point 3 in the original review), whilst I agree that the stoichiometry of peptide phosphorylation by Ulk1 is not meaningful, I think it would be very informative to include the stoichiometry of phosphorylation of S108 in the AMPK complex by Ulk1.
3. In the new Figure 7a, quantification of pT172 and pACC should be included.
4. Although it is very weak, there appears to be a signal for pACC in the a1/2 dKO iMEFs shown in Suppl. Fig. 5. Is it known what this represents (perhaps AMPK is not the exclusive ACC-Ser79 kinase)? Also, the labelling for the blot should be alpha (not alpha1)?

Reviewer #3 (Remarks to the Author):

The authors have detailed their efforts to experimentally address some of this reviewer's relatively minimal concerns. It is understood that many of these are technically difficult and were not able to be cleanly interpreted due to alterations in phosphatases or other modulators of Ser108 during transformation, etc.

The new data using a T172A AMPKa mutant introduced into the AMPKa1/a2 DKO cells will be of interest to many so with this, the manuscript should be suitable for publication without further revision in this reviewer's opinion.

Reviewer #4 (Remarks to the Author):

Dear Editor

Thanks for giving me the opportunity to stand-in in this review process.

The original referee #2 clearly looks for more physiological relevant insights to an otherwise fascinated "story" on alternative activity regulation of AMPK. The authors do not provide this. Cellular localization is judged to be outside the scope of the MS. Yes, but very relevant.

The list of alternative kinases is only partially provided. A demand for the full list seems to be a valid request since the ULK dKO induces a paradoxically elevation in basal S108 phosphorylation. The authors suggest that PPase activity is decreased – but this could be other kinases as well. Physiological relevant impact may be revealed by investigating the pS108-activated AMPK downstream targets to that of pThr172 activated AMPK. This would be a very important step to take

Metformin is judged to work a-like phenformin (via respiration inhibition) – this seems as a plausible explanation but of course if it is not measured we cannot know.

In conclusion;

This is a nice MS with loads of new molecular insights. However, evidence for the physiological relevance is missing in the present version. I would suggest that the whole list of kinases is

provided. It would also be very insightful if differential AMPK targets could be provided. This latter point may also be judge by the comments from ref 1 and 3 who seem to have had similar views.

Reviewer #1:

We thank the Reviewer for his/her constructive comments. We have addressed concerns or suggestions individually below:

The authors have made a number of changes to the manuscript in order to respond to my initial comments. These have improved the manuscript, although I think it is still unclear what, if any, is the physiological significance of the findings.

In our previous response we provided detailed explanation of the technical challenges associated with measuring the contribution ULK-mediated phosphorylation of Ser108 makes to global AMPK signalling. Nevertheless, we have employed quantitative proteomic and phosphoproteomic analyses to interrogate the physiological impact of Ser108 phosphorylation *per se*. These studies are described in new Fig 2 and Suppl. Fig. 2. We transduced β 1/2 dKO MEFs with either β 1 mutants S108A (non-phosphorylated) or S108E (an effective phosphomimetic, see Scott et al Chem. Biol. 21, 619-627 (2014)) and treated with phenformin to induce an energy stress. The study is limited in that we do not know whether metabolite(s) acting at the ADaM site (which is stabilized in the S108E mutant, but not S108A) are elevated under these conditions and able to significantly trigger AMPK activity.

We detected 15 phosphoproteins with significantly altered phosphorylation profile in S108E cells vs S108A cells. Importantly, there were no detectable differences in AMPK expression or phenformin-induced Thr172 phosphorylation between cell types. We attribute these differences to AMPK signalling through the ADaM site, providing insight into physiological relevance of Ser108 phosphorylation. We confirmed increased phosphorylation on one of these sites (PAK2-S141) by Western blot. 4 of these perturbed cellular phosphoproteins are closely linked to cell cycle regulation (PAK2, Bin1, CDN1 and TB182); collectively Ingenuity Pathway Analysis (Qiagen) identified 'Cell cycle, Connective Tissue Development and Function, Cellular Movement' as one of the top associated networks. These findings provide additional depth to the manuscript, and form a backdrop for future studies aimed at delineating the cellular signalling mechanisms specific to β 1-Ser108 phosphorylation. We have included discussion points based on this study, that warranted slight re-arrangement of discussion sections. We have removed other non-essential discussion points to prevent an increase in overall length.

The mass spectrometry proteomics data have been deposited to the ProteomeXchange Consortium via the PRIDE partner repository (<http://www.ebi.ac.uk/pride>). The Reviewer is welcome to access the dataset using the following details:

Project Name: Quantitative global and phosphoproteomic analysis to define the cellular role of AMPK β 1-Ser108 phosphorylation

Project accession: PXD006365

Project DOI: Not applicable

Reviewer account details:

Username: reviewer69753@ebi.ac.uk

Password: V2k5ysGn

There are a couple of points arising in the revised manuscript that the authors should address.

1. In the new version of Figure 2d, a blot for the parental cells (or empty lenti virus control) should be included.

An immunoblot for AMPK α , β 1 and pACC in α 1/2 dKO parental cells (untransduced) is already presented in Fig. 3c, lane 1. This panel was Fig 2c in the previous version.

2. For Figure 3 (and referring back to point 3 in the original review), whilst I agree that the stoichiometry of peptide phosphorylation by Ulk1 is not meaningful, I think it would be very informative to include the stoichiometry of phosphorylation of S108 in the AMPK complex by Ulk1.

We have now included % Ser108 phosphorylation of ULK1 and ULK2 treated heterotrimer in Fig 4c (old Fig 3c), based on the ToF spectrum of CaMKK2-treated AMPK β -subunit, blotted in lane 1.

3. In the new Figure 7a, quantification of pT172 and pACC should be included.

We have added a graph presenting fold-increase in pACC and pT172, relative to basal phosphorylation in WT α 2 transduced cells (now Figure 8).

4. Although it is very weak, there appears to be a signal for pACC in the α 1/2 dKO iMEFs shown in Suppl. Fig. 5. Is it known what this represents (perhaps AMPK is not the exclusive ACC-Ser79 kinase)?

We commend the Reviewer on their excellent eyesight! We attribute the very faint band in lane 2 (and partially lane 3) to non-specific signal from the 2° antibody. This is not present in immunoblots from the other 2 replicates, one of which we now present in Suppl. Fig. 7 (previous Suppl. Fig. 5). Our conclusion that AMPK is the sole ACC-Ser79 kinase is supported by independent studies using AMPK α 1/2 or β 1/2 double KO cells (e.g. Laderoute et al (2006) *Mol. Cell. Biol.* 26(14):5336; Goransson et al (2007) *J. Biol. Chem.* 282:32549; Johans et al (2016) *Nat. Commun.* 7:10856; Willows et al, (2017) *Biochem. J.* doi: 10.1042/BCJ20170046).

5. Also, the labelling for the blot should be alpha (not alpha1)?

Thank you, this has been corrected.

Reviewer #3:

The authors have detailed their efforts to experimentally address some of this reviewer's relatively minimal concerns. It is understood that many of these are technically difficult and were not able to be cleanly interpreted due to alterations in phosphatases or other modulators of Ser108 during transformation, etc.

The new data using a T172A AMPK α mutant introduced into the AMPK α 1/ α 2 DKO cells will be of interest to many so with this, the manuscript should be suitable for

publication without further revision in this reviewer's opinion.

We thank the Reviewer for acknowledging the technical difficulties associated with demonstrating the impact of ULK-mediated Ser108 phosphorylation.

The Reviewer may be interested to know that we have now employed quantitative proteomic and phosphoproteomic analyses to interrogate the physiological impact of Ser108 phosphorylation *per se*. These studies are described in new Fig 2 and Suppl. Fig. 2. We transduced $\beta 1/2$ dKO MEFs with either $\beta 1$ mutants S108A (non-phosphorylated) or S108E (an effective phosphomimetic, see Scott et al Chem. Biol. 21, 619-627 (2014)) and treated with phenformin to induce an energy stress. The study is limited in that we do not know whether metabolite(s) acting at the ADaM site (which is stabilized in the S108E mutant, but not S108A) are elevated under these conditions and able to significantly trigger AMPK activity.

We detected 15 phosphoproteins with significantly altered phosphorylation profile in S108E cells vs S108A cells. Importantly, there were no detectable differences in AMPK expression or phenformin-induced Thr172 phosphorylation between cell types. We attribute these differences to AMPK signalling through the ADaM site, providing insight into physiological relevance to Ser108 phosphorylation. We confirmed increased phosphorylation on one of these sites (PAK2-S141) by Western blot. 4 of these perturbed cellular phosphoproteins are closely linked to cell cycle regulation (PAK2, Bin1, CDN1 and TB182); collectively Ingenuity Pathway Analysis (Qiagen) identified 'Cell cycle, Connective Tissue Development and Function, Cellular Movement' as one of the top associated networks. These findings provide additional depth to the manuscript, and form a backdrop for future studies aimed at delineating the cellular signalling mechanisms specific to $\beta 1$ -Ser108 phosphorylation. We have included discussion points based on this study, that warranted slight re-arrangement of discussion sections. We have removed other non-essential discussion points to prevent an increase in overall length.

The mass spectrometry proteomics data have been deposited to the ProteomeXchange Consortium via the PRIDE partner repository (<http://www.ebi.ac.uk/pride>). The Reviewer is welcome to access the dataset using the following details:

Project Name: Quantitative global and phosphoproteomic analysis to define the cellular role of AMPK $\beta 1$ -Ser108 phosphorylation

Project accession: PXD006365

Project DOI: Not applicable

Reviewer account details:

Username: reviewer69753@ebi.ac.uk

Password: V2k5ysGn

Reviewer #4:

Dear Editor

Thanks for giving me the opportunity to stand-in in this review process.

The original referee #2 clearly looks for more physiological relevant insights to an otherwise fascinated "story" on alternative activity regulation of AMPK. The authors do not provide this.

We thank the Reviewer for stepping in for Reviewer 2 and reiterating his/her comments. We have addressed concerns or suggestions individually below:

1. Cellular localization is judged to be outside the scope of the MS. Yes, but very relevant.

We agree, this is relevant and is currently the focus of a new study.

2. The list of alternative kinases is only partially provided. A demand for the full list seems to be a valid request since the ULK dKO induces a paradoxically elevation in basal S108 phosphorylation. The authors suggest that PPase activity is decreased – but this could be other kinases as well.

The list of top S108 kinase hits identified in our screen (>10% activity rel. to AMPK $\alpha 1\beta 2\gamma 1$) is now included in new Suppl. Fig. 3.

3. Physiological relevant impact may be reveal by investigating the pS108-activated AMPK downstream targets to that of pThr172 activated AMPK. This would be a very important step to take.

In our previous response we provided detailed explanation of the technical challenges associated with measuring the contribution ULK-mediated phosphorylation of Ser108 makes to global AMPK signalling. Nevertheless, we have employed quantitative proteomic and phosphoproteomic analyses to interrogate the physiological impact of Ser108 phosphorylation *per se*. These studies are described in new Fig 2 and Suppl. Fig. 2. We transduced $\beta 1/2$ dKO MEFs with either $\beta 1$ mutants S108A (non-phosphorylated) or S108E (an effective phosphomimetic, see Scott et al Chem. Biol. 21, 619-627 (2014)) and treated with phenformin to induce an energy stress. The study is limited in that we do not know whether metabolite(s) acting at the ADaM site (which is stabilized in the S108E mutant, but not S108A) are elevated under these conditions and able to significantly trigger AMPK activity.

We detected 15 phosphoproteins with significantly altered phosphorylation profile in S108E cells vs S108A cells. Importantly, there were no detectable differences in AMPK expression or phenformin-induced Thr172 phosphorylation between cell types. We attribute these differences to AMPK signalling through the ADaM site, providing insight into physiological relevance to Ser108 phosphorylation. We confirmed increased phosphorylation on one of these sites (PAK2-S141) by Western blot. 4 of these perturbed cellular phosphoproteins are closely linked to cell cycle regulation (PAK2, Bin1, CDN1 and TB182); collectively Ingenuity Pathway Analysis (Qiagen) identified ‘Cell cycle, Connective Tissue Development and Function, Cellular Movement’ as one of the top associated networks. These findings provide additional depth to the manuscript, and form a backdrop for future studies aimed at delineating the cellular signalling mechanisms specific to $\beta 1$ -Ser108 phosphorylation. We have included discussion points based on this study, that warranted slight re-arrangement of discussion sections. We have removed other non-essential discussion points to prevent an increase in overall length.

The mass spectrometry proteomics data have been deposited to the ProteomeXchange Consortium via the PRIDE partner repository (<http://www.ebi.ac.uk/pride>). The Reviewer is welcome to access the dataset using the following details:

Project Name: Quantitative global and phosphoproteomic analysis to define the cellular role of AMPK β 1-Ser108 phosphorylation

Project accession: PXD006365

Project DOI: Not applicable

Reviewer account details:

Username: reviewer69753@ebi.ac.uk

Password: V2k5ysGn

4. Metformin is judged to work a-like phenformin (via respiration inhibition) – this seems as a plausible explanation but of course if it is not measured we cannot know.

The inhibitory effects of metformin on mitochondrial respiration are well documented (eg Andrzejewski et al, Cancer Metab. 2014). We have investigated the effect of metformin on Ser108 phosphorylation in HEKs. HEK293T cells expressing GST-tagged kinase-dead (KI)-AMPK α 1 β 1 γ 1 were treated with metformin (70 μ M or 1 mM) or phenformin (1 mM) for 1 h. This is the same incubation period used for phenformin treatment in Fig 3b. The previous study referred to by Reviewer 2 (Zhang et al, Cell Metab. 2016) used a 12 h metformin incubation, which we deem excessive for the mechanism we are investigating here.

We did not detect pSer108 on GST-purified AMPK, nor did we detect increase in AMP/ATP ratios, from metformin treated cells, as shown below. These results are consistent with metformin being a weaker inhibitor of mitochondrial respiration than phenformin, and lack of OCT1 transporter in these cells, shown to be required for metformin uptake in hepatocytes. We have not included this data in the manuscript, but will do so at the discretion of the Editor.

In conclusion;

This is a nice MS with loads of new molecular insights. However, evidence for the physiological relevance is missing in the present version. I would suggest that the whole list of kinases is provided. It would also be very insightful if differential AMPK targets could be provided. This latter point may also be judge by the comments from ref 1 and 3 who seem to have had similar views.

REVIEWERS' COMMENTS:

Reviewer #1 (Remarks to the Author):

The authors have made a number of changes in response to the second round of comments and these have improved the manuscript.

Ultimately, the physiological significance of these findings will require additional studies that are outside the scope of the current study.

Reviewer #4 (Remarks to the Author):

I have no further comments. As said previously - this study/MS reveals new interesting aspects to AMPK regulation.

Reviewer 1:

Ultimately, the physiological significance of these findings will require additional studies that are outside the scope of the current study.

We have already included text indicating the need for further study to delineate the roles pS108 plays in AMPK signalling (line 436)